. Pathogens

# Characterization of a novel cell wall-associated nucleotidase of *Enterococcus faecalis* that degrades extracellular c-di-AMP

Adriana G. Morales Rivera[1], Anju Bala[1], Leila G. Casella[1], Debra N. Brunson[1], Ellsa Wongso[2], Aria Patel[1], Ilka E. Cuvilly[1], Alejandro R. Walker[1], Shannon M. Wallet[1], Ana L. Flores-Mireles[2], José A. Lemos[1]*

**1** Department of Oral Biology, University of Florida College of Dentistry, Gainesville, Florida, United States of America, **2** Department of Biological Sciences, University of Notre Dame, Notre Dame, Indiana, United States of America

* jlemos@dental.ufl.edu

## Abstract

*Enterococcus faecalis* is a prolific opportunistic pathogen responsible for a range of life-threatening infections, notorious for its ability to withstand environmental stressors. Second messengers are small molecules that relay signals in response to stimuli and are thought to be crucial for bacteria like *E. faecalis* to modulate their adaptation to stress. The second messenger nucleotide c-di-AMP has emerged as an essential bacterial signaling molecule due to its impact on physiological processes, including adaptation to osmotic stress, cell wall homeostasis, antibiotic tolerance, and virulence. In addition, c-di-AMP is a pathogen-associated molecular pattern (PAMP) molecule that can trigger a potent stimulation of the host immune system. In previous work, we identified and characterized the enzymes responsible for the synthesis and degradation of intracellular c-di-AMP in *E. faecalis*, demonstrating that maintenance of c-di-AMP homeostasis is vital for its fitness and virulence. In addition to the intracellular enzymes that regulate c-di-AMP levels, a small number of bacteria encode surface-associated nucleotidases that cleave extracellular c-di-AMP and are potentially associated with immune evasion. Here, we characterize a novel and unique cell wall-anchored nucleotidase, termed EecP (*E. faecalis* extracellular c-di-AMP phosphodiesterase), which features duplicated catalytic domains and degrades extracellular c-di-AMP. Through competition experiments, we show that EecP likely uses c-di-GMP, and to a lesser extent AMP, as additional substrates. While a Δ*eecP* strain failed to display relevant phenotypes under most *in vitro* conditions, it exhibited increased susceptibility to killing by phagocytic cells, a phenotype at least partly associated with cGAS-STING immune signaling. NanoString analysis revealed distinct innate immune signatures in murine macrophages infected with the parent strain OG1RF or Δ*eecP*, uncovering differential expression of host targets known to be impacted by c-di-AMP, as well as novel targets. Using two murine infection models, we show that the impact of *eecP* deletion and the consequent buildup of extracellular

**Data availability statement:** All data are in the manuscript and/or Supporting information files.

**Funding:** This study was supported by NIH-NIAID R01 AI172179 to J.A.L. D.N.B. was supported by NIH-NIDCR Training Grant T90 DE021990. A.G.M.R. was supported by the Florida Education Fund McKnight Doctoral Fellowship. The funders had no role in study design, data collection and analysis, decision to publish, or preparation of the manuscript.

**Competing interests:** The authors have declared that no competing interests exist.

c-di-AMP on *E. faecalis* pathogenesis might depend on the site of infection. Notably, disseminated infection was more severe in mice infected with Δ*eecP,* suggesting that extracellular c-di-AMP influences infection outcomes, likely through modulation of host immune responses.

---

## Author summary

*Enterococcus faecalis* is a major opportunistic pathogen and a leading cause of several life-threatening hospital-associated infections. The ability of *E. faecalis* to cause disease is closely tied to its capacity to withstand stress. Second messengers are small molecules that allow bacteria to respond to stimuli, including stressors, in their environment. Cyclic di-AMP is a bacterial second messenger nucleotide that regulates essential cellular processes and plays key roles in bacterial pathogenesis and activation of host immune responses. We previously characterized the enzymes responsible for the synthesis and degradation of c-di-AMP in *E. faecalis*, demonstrating that this signaling molecule is crucial for bacterial fitness and virulence. In this study, we describe the characterization of EecP, a novel cell wall-associated enzyme that rapidly degrades bacterial cyclic dinucleotides extracellularly. Our findings identify EecP as a novel immune-modulating factor that can influence host responses and infection outcomes.

## Introduction

*Enterococcus faecalis* is a Gram-positive facultative anaerobe and commensal inhabitant of the gastrointestinal tract of humans and animals. Notorious for its prevalence as a major healthcare-associated opportunistic pathogen, *E. faecalis* was responsible for an estimated toll of over 200,000 worldwide deaths in 2019 alone [1]. Generally considered a low-grade pathogen due to its limited repertoire of tissue-damaging virulence factors, the pathogenic potential of *E. faecalis* largely stems from its ability to form robust biofilms on host tissues and medical devices, evade the immune system, and survive under various stressful conditions. These factors collectively enable its persistence in hospital environments and contribute to opportunistic infections such as catheter-associated urinary tract infections (CAUTIs), intra-abdominal and soft tissue infections, and central line-associated bloodstream infections [2,3]. Moreover, *E. faecalis* possesses intrinsic and acquired resistance to multiple antibiotics, further complicating treatment and worsening prognosis [4].

Second messengers are small signaling molecules that enable bacteria to respond to environmental stressors and modulate various aspects of bacterial physiology and virulence. Since its serendipitous discovery in the crystal structure of the DNA integrity scanning protein DisA from *Thermotoga maritima* in 2008 [5], bis-(3′-5′)-cyclic dimeric adenosine monophosphate (cyclic di-AMP or c-di-AMP) has emerged as an essential nucleotide second messenger for bacteria, required for regulating

numerous functions, including osmotic adaptation, DNA repair, cell wall homeostasis, antibiotic tolerance, and virulence [6–12]. Cyclic di-AMP is synthesized from two adenosine triphosphate (ATP) molecules by diadenylate cyclase (DAC) enzymes and degraded to phosphoadenylyl adenosine (pApA) or adenosine monophosphate (AMP) by enzymes collectively known as phosphodiesterases (PDE). As a ligand, c-di-AMP binds allosterically to effectors and positively or negatively alters their activity [6–12]. To date, over twenty targets have been identified to be under c-di-AMP allosteric control, including membrane-associated transporters, signal transduction proteins, metabolic enzymes, regulatory proteins, and riboswitches [6–12].

While essential for cellular functions, bacteria must tightly control intracellular c-di-AMP pools by modulating the activities of DAC and PDE enzymes and, arguably, through active secretion mechanisms [13]. Bacterial strains that cannot synthesize c-di-AMP are frequently conditionally viable, with several examples indicating that c-di-AMP production is essential for bacterial virulence [14–16]. At the same time, PDE mutants that accumulate intracellular c-di-AMP also display adverse phenotypes, from altered stress responses to attenuated virulence in mouse models [17–19]. Previously, our group characterized the enzymes that synthesize (CdaA) and degrade (DhhP and GdpP) c-di-AMP in *E. faecalis* [15]. The key finding of this study was that disruption of c-di-AMP homeostasis, either the intracellular accumulation observed in strains lacking one or both PDEs (Δ*gdpP*, Δ*dhhP*, and Δ*dhhP* Δ*gdpP*) or the absence of c-di-AMP observed in the Δ*cdaA* strain, drastically impaired *E. faecalis* fitness and virulence [15].

Beyond its role in bacterial physiology, c-di-AMP is also a major pathogen-associated molecular pattern (PAMP) molecule recognized by host immune and tissue cells [9,20–22]. Specifically, prior studies have shown that c-di-AMP is recognized by at least four host cell receptors, namely STING (Stimulator of Interferon Genes) [20,21,23], DDX41 (DEAD-box RNA Helicase 41) [24], RECON (Reductase Controlling NF-κB or AKR1C13) [25], and ERAdP (Endoplasmic Reticulum Membrane Adaptor) [26]. These receptors activate various innate immune signaling pathways that promote the production of cytokines that modulate immune responses against bacteria. As such, c-di-AMP sensing by the host is thought to play a key role in antibacterial responses that work to limit infection progression.

In addition to cytoplasmic PDEs, a few bacteria have been shown to encode surface-associated enzymes, termed extracellular phosphodiesterases (ePDE), that hydrolyze extracellular c-di-AMP. The first thorough description of an ePDE that degrades c-di-AMP was in the human pathogen *Streptococcus agalactiae* with the identification of the cell wall-anchored enzyme CdnP [20]. In this study, CdnP was shown to degrade extracellular c-di-AMP, dampening activation of the STING-mediated response, leading the authors to propose that CdnP serves as an immune evasion strategy for *S. agalactiae* [20]. A second c-di-AMP ePDE, SntA, identified in the zoonotic pathogen *Streptococcus suis*, was initially characterized as a cell wall-anchored heme-binding protein that interferes with complement-mediated killing [27,28]. Biochemical studies confirmed that SntA can efficiently degrade c-di-AMP as well as other nucleotides, including 3′-AMP, pApA, and 2′3′-cAMP [29]. Moreover, SntA was proposed to work alongside other extracellular nucleotidases of *S. suis* to degrade adenine nucleotides and generate adenosine (Ado), which can impact bacterial survival and virulence [30]. More recently, an ePDE termed CpdB, capable of degrading c-di-AMP, 2′3′-cGAMP, among other nucleotides, was identified in *Bacillus anthracis* [31]. Loss of CpdB led to enhanced colonization despite a reduction in *B. anthracis* virulence in systemic silkworm and mouse infection models [31].

Through BLASTp searches and structural predictions, we identified a surface-anchored nucleotidase in the *E. faecalis* reference strain OG1RF [32] and termed it EecP for *E. faecalis* extracellular c-di-AMP phosphodiesterase. Deletion of *eecP* led to a significant accumulation of c-di-AMP in the extracellular space, confirming its role in nucleotide degradation. Using a site-directed mutagenesis approach, we identified a histidyl residue within an asparagine-histidine-glutamate (NHE) motif that is crucial for c-di-AMP degradation by EecP. Competition assays revealed that EecP can likely degrade c-di-GMP, and to a lesser degree, AMP, suggesting it may also have the capacity to modulate interspecies and interkingdom nucleotide signaling. While inactivation of *eecP* did not alter the survival of *E. faecalis* under *in vitro* conditions typically associated with c-di-AMP regulation, the Δ*eecP* strain showed increased sensitivity to macrophage killing,

a phenotype that was linked, at least in part, to c-di-AMP activation of the cGAS-STING signaling pathway. NanoString multiplex technology was used to analyze gene expression of macrophages infected with the parent or Δ*eecP* strain. The results indicated that each strain elicited distinct innate immune responses that included host targets known to be impacted by c-di-AMP, as well as novel targets. Using two mouse infection models, we demonstrate that the impact of *eecP* inactivation and the resulting accumulation of extracellular c-di-AMP on *E. faecalis* pathogenesis may vary depending on the site of infection. Notably, systemic dissemination was more severe in animals infected with the Δ*eecP* strain compared to those infected with the parent strain. Together, these findings identify EecP as a novel surface-associated immune-modulating factor capable of shaping infection outcomes.

## Results

### Identification of a cell surface-associated phosphodiesterase that is unique to *E. faecalis*

Using *in silico* analyses, we identified a putative cell wall-anchored phosphodiesterase in the genome of *E. faecalis* OG1RF, designated OG1RF_RS00285, herein referred to as EecP (*E. faecalis* extracellular c-di-AMP phosphodiesterase). EecP contains an N-terminal secretion signal and is predicted to be anchored to the peptidoglycan cell wall via a C-terminal LPxTG motif (Fig 1A). Unlike other c-di-AMP-specific ePDEs, which typically consist of a single metallophosphoesterase-nucleotidase (MP-NT) two-domain structure, EecP features duplicated MP-NT structures (Fig 1A–1B). A structure prediction of the full-length EecP can be seen highlighted in teal in Fig 1B. The structure is characterized by two independent, organized MP-NT domain pairs and is flanked by two intrinsically disordered regions, which harbor the secretion signal at the N-terminus and the LPxTG cell wall sortase signal at the C-terminus. The C-terminus disordered region is characterized by a proline-rich section with proline-aspartate-proline-lysine (PDPK) repeats. Although the two MP-NT domain pairs of EecP (MP-NT/D1 and MP-NT/D2) exhibit low amino acid identity (~24%), they are predicted to be structurally similar, with a root mean square deviation (RMSD) of 4.56 and a template modeling (TM) score of 0.72 (Fig 1B–1C).

The activity of bacterial nucleotidases with MP-NT domain pairs was previously described in the studies of *Escherichia coli* UshA and CpdB and *S. suis* SntA [33–35]. Collectively, these studies suggest that the MP domain (Pfam: PF00149) is responsible for enzymatic activity, while the NT domain (Pfam: PF02872) mediates substrate binding and specificity [36]. Furthermore, the enzymatic activity of MP domains is attributed to a key histidyl (H) residue within a conserved 'NHE' motif [20,35–37]. These conserved histidyl residues can be identified in the MP-NT/D1 (H203) and MP-NT/D2 (H758) domain pairs of EecP, suggesting that both MP-NT domain structures of EecP are catalytically active (Figs 1D–1F and S1). In the *S. suis* SntA, two tyrosine residues within 'DxYxYxN' and 'NNYR' motifs (Y530 and Y633) in the NT domain were predicted to bind substrates by forming a sandwich-like structure and shown experimentally to impact substrate binding efficiency and specificity [29]. While both tyrosine residues are conserved in MP-NT/D1 (Y491 and Y580), these residues are replaced by phenylalanines in MP-NT/D2 (F1043 and F1159) (Fig 1D–1F). The presence of these key residues with similar side chain structures in EecP MP-NT/D1 and MP-NT/D2 suggests that there may be a conserved function but unique specificity to these duplicated domain structures.

Because nucleotidases with MP-NT domain pair structures have been previously identified in both Gram-positive and Gram-negative bacteria, we performed BLASTp searches using EecP as the query sequence in NCBI and BV-BRC databases to identify homologs, with particular emphasis on cell wall-anchored proteins [38]. We found that homologs with duplicated MP-NT domain structures are present in only a few closely related organisms, including *Enterococcus caccae*, *Enterococcus haemoperoxidus,* and *Vagococcus lutrae* (Fig 2). Despite the conservation of EecP across *E. faecalis* strains, no homologs were identified in *E. faecium* genomes, the second most prevalent human-associated enterococci. Of interest, the Gram-positive model organism *B. subtilis* is one of the few non-enterococcal species harboring a large surface-anchored nucleotidase, named YfkN, with a duplicated MP-NT domain structure. YfkN has been shown to have phosphodiesterase activity against 2'3'-cyclic nucleotides and 5'-mononucleotides [39]. Contrary to proteins containing

PLOS Pathogens

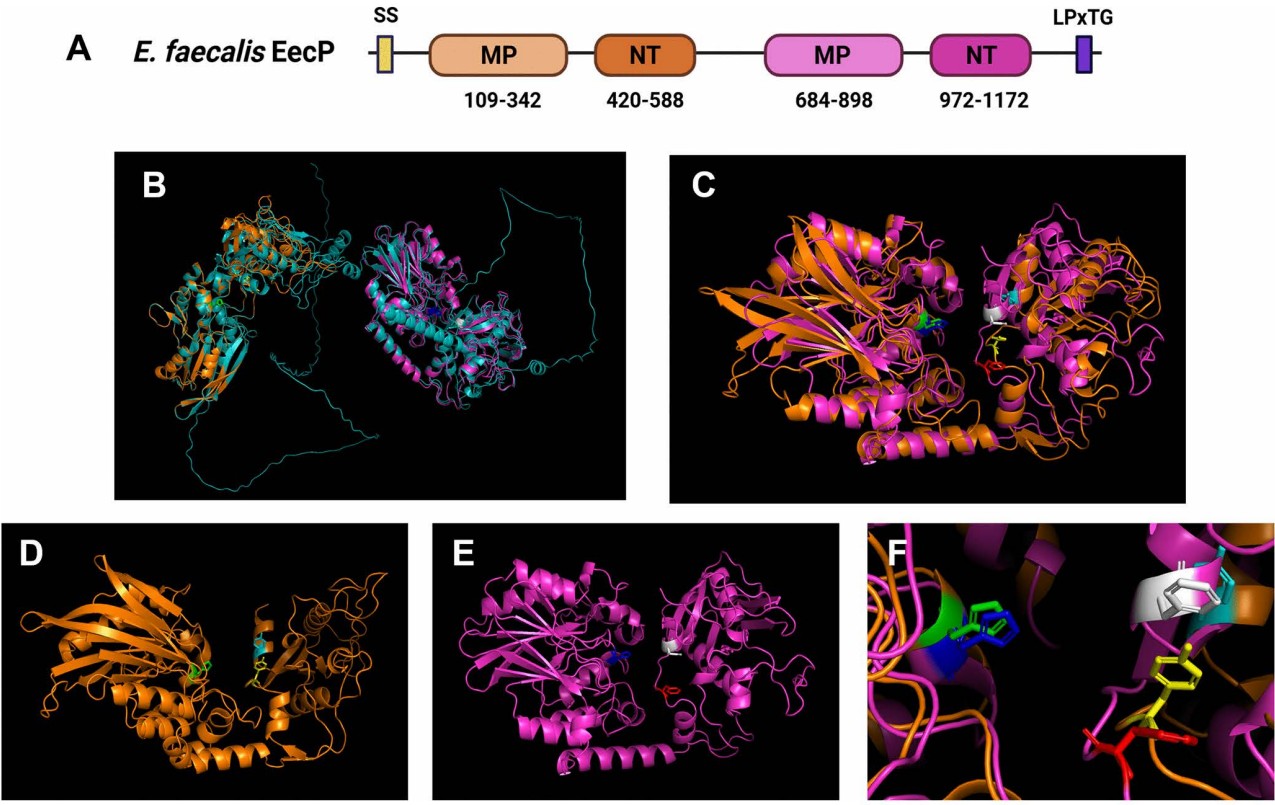

**Fig 1. EecP is an *E. faecalis* extracellular c-di-AMP phosphodiesterase. (A)** Domain structure of EecP, SS: secretion signal, MP: metallophosphoe-sterase, NT: 5′-nucleotidase, LPxTG: cell-surface anchoring motif. **(B)** Structure prediction of *E. faecalis* EecP (OG1RF_RS00285) showing two distinct structures containing metallophosphoesterase (MP) and nucleotidase (NT) domain pairs (MP-NT/D1 and MP-NT/D2). MP-NT/D1 is highlighted in orange, MP-NT/D2 in magenta, and the full-length structure in teal. **(C)** MP-NT/D1 and MP-NT/D2 domain pairs superimposed on each other. **(D)** MP-NT/D1 structure highlighting the conserved histidyl (H203 in green) and tyrosine residues (Y491 in yellow and Y580 in cyan). **(E)** MP-NT/D2 structure highlighting the conserved histidyl (H758 in dark blue) and phenylalanine residues (F1043 in red and F1159 in white). **(F)** Close-up of H203 (green), Y491 (yellow), and Y580 (cyan) of MP-NT/D1 and H758 (dark blue), F1043 (red), and F1159 (white) of MP-NT/D2. All structure predictions were obtained using AlphaFold3 via AlphaFold Server, and all visualizations were generated with PyMol.

duplicated MP-NT domains, cell wall-anchored nucleotidases with a single MP-NT domain structure are more prevalent in bacteria, including important pathogens such as *S. agalactiae*, *Streptococcus pyogenes*, *Staphylococcus aureus*, *Listeria monocytogenes*, and *B. anthracis* (Fig 2).

### *E. faecalis* rapidly degrades extracellular c-di-AMP

Because attempts to purify various full-length and truncated versions of EecP using different expression systems were unsuccessful, we developed a thin-layer chromatography (TLC)-based assay (see methods for details) to monitor the degradation of radiolabeled c-di-AMP using live bacterial cell suspensions. To track product formation, we used radiola-beled ATP and c-di-AMP and resolved the c-di-AMP degradation products formed by purified *E. faecalis* DhhP (intracellu-lar PDE) as standard controls (S2A Fig). As results from TLC analysis are mainly qualitative, to ensure rigor, we monitored c-di-AMP degradation in the supernatants of the *E. faecalis* reference strains OG1RF and V583. As shown in Fig 3, both strains rapidly degraded c-di-AMP, generating byproducts corresponding to pApA, AMP, and inorganic phosphate (Pi). To rule out spontaneous degradation of c-di-AMP over time, radiolabeled c-di-AMP was kept in the same buffer used to

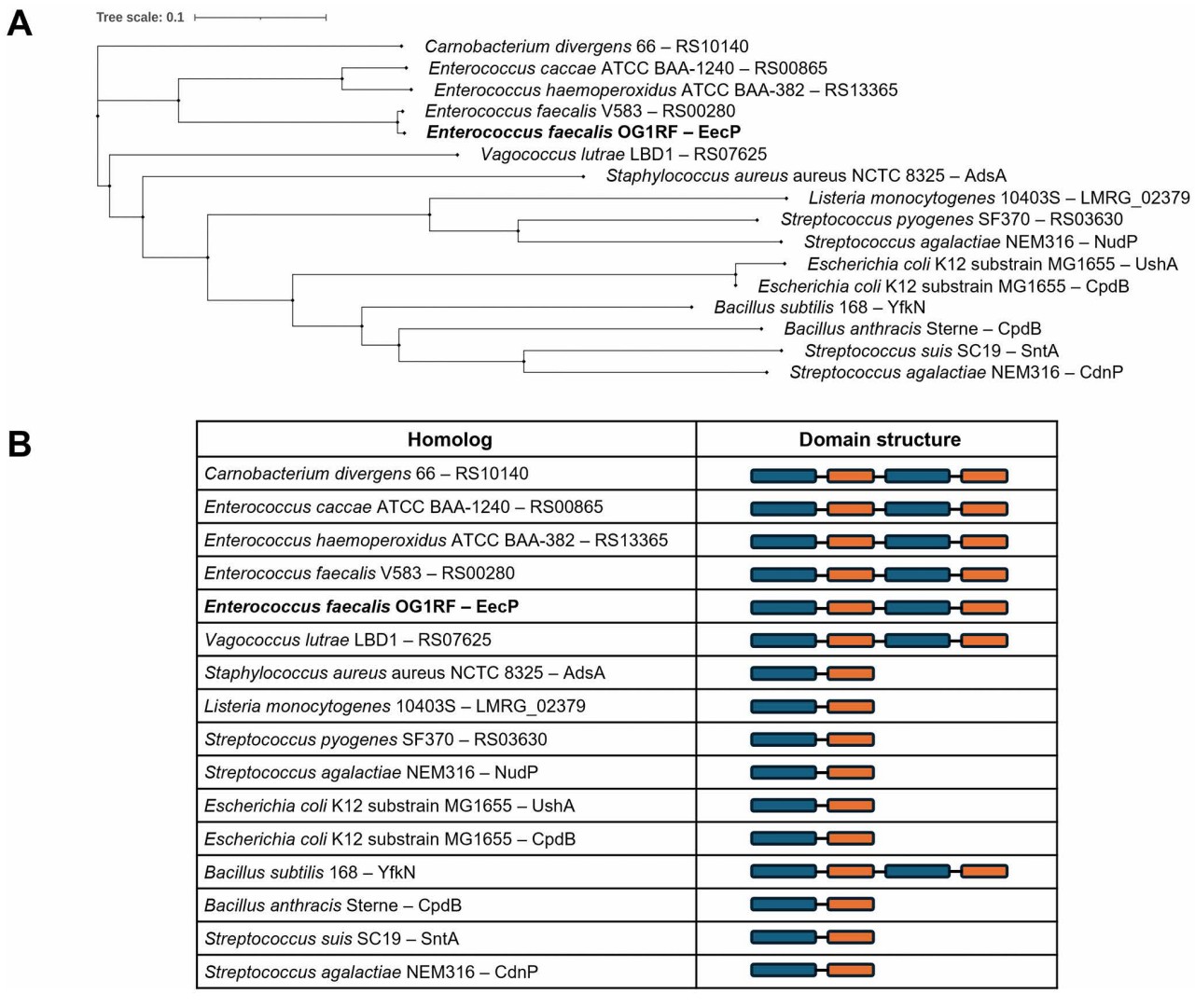

**Fig 2. Phylogenetic analysis of EecP homologs. (A)** Phylogenetic tree was constructed using multiple sequence alignments of putative homologs using EMBL-EBI Clustal Omega and iTOL. **(B)** Domain structures of EecP homologs. BLASTp searches using EecP as the query sequence in NCBI and BV-BRC were used to identify homologs in Gram-positive and Gram-negative bacteria. Protein sequences were obtained from genomes of representative strains indicated for each species available in the NCBI or BioCyc databases.

resuspend live cells for up to 6 hours without observing any detectable loss in the intensity of the c-di-AMP spot (S2B Fig). These results point to the capability of *E. faecalis* to rapidly degrade extracellular c-di-AMP into pApA and AMP at the cell surface interface. Comparable results were obtained using *S. agalactiae* strains that encode the c-di-AMP ectonucleotidase CdnP (Fig 3).

## Loss of *eecP* leads to extracellular accumulation of c-di-AMP

To investigate the role of EecP in c-di-AMP degradation, we used a markerless in-frame strategy [40] to generate a clean *eecP* gene deletion (Δ*eecP*) in the reference strain OG1RF and used the cCF10 pheromone-inducible vector pCIE to genetically complement the mutant strain (Δ*eecP* pCIE::*eecP*) [41]. To verify cell wall localization of EecP, cell

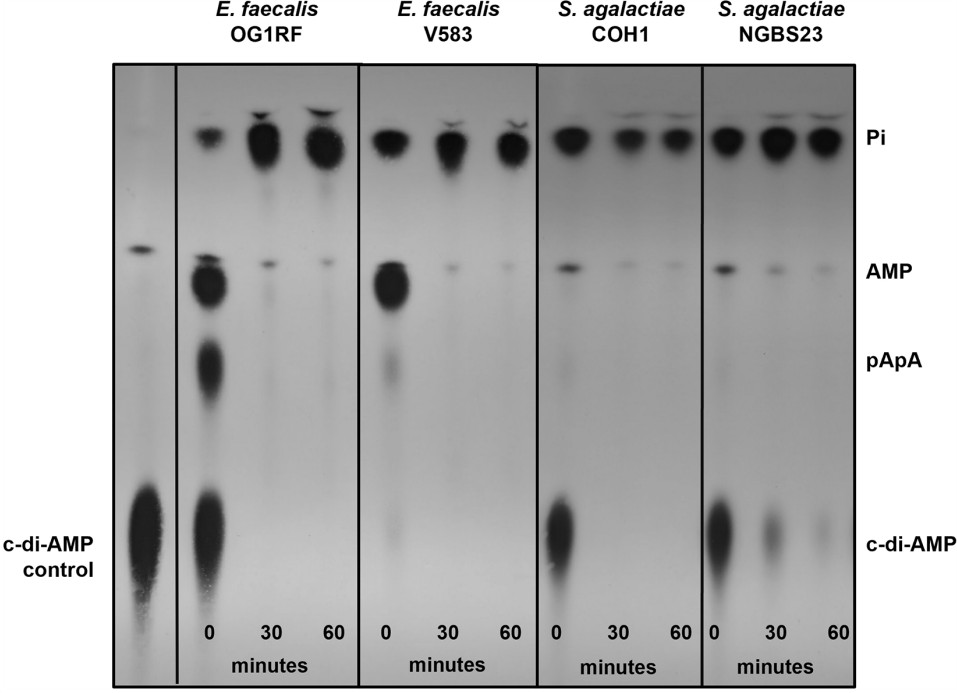

**Fig 3. Degradation of extracellular c-di-AMP in *E. faecalis* and *S. agalactiae* strains.** Mid-log cultures grown in THB (~$OD_{600}$ 0.4) and suspended in 50 mM Tris-Cl containing 5 mM $MnCl_2$ were added [$^{32}$P]-c-di-AMP and sampled over time for c-di-AMP degradation. Reaction aliquots were collected at the indicated time points and inactivated by boiling before spotting on a PEI-cellulose plate for TLC separation. TLC image is a representative of experiment conducted at least two times with independent biological replicates.

wall-associated proteins from OG1RF and Δ*eecP* strains were isolated from late-log phase cultures using enzymatic digestion with lysozyme and separated by SDS-PAGE. SYPRO Ruby staining revealed a putative band for EecP around ~150 kD in OG1RF that was absent in the Δ*eecP* strain (S3 Fig). The identity of the band was verified to be EecP by mass spectrometry analysis. Next, we used a commercially available ELISA to quantify intra- and extracellular c-di-AMP pools in the OG1RF, Δ*eecP,* and Δ*eecP* pCIE::*eecP* strains during different phases of growth *in vitro*. Briefly, cultures were grown in chemically defined media (CDM), and both cell lysates and supernatants were collected at early-log ($OD_{600}$ 0.4), mid-log ($OD_{600}$ 0.6), and late-log ($OD_{600}$ 0.8) phases for intra- and extracellular c-di-AMP quantifications, respectively [42,43]. While there were no significant differences in intracellular c-di-AMP pools between the strains, extracellular c-di-AMP levels were significantly higher in Δ*eecP* supernatants when compared to OG1RF supernatants throughout the different growth phases (Fig 4A–4B). Genetic complementation of the Δ*eecP* strain restored extracellular c-di-AMP to parent strain levels (Fig 4C).

To validate the ELISA results and obtain insights into the c-di-AMP degradation products of EecP, we monitored extracellular [$^{32}$P]-c-di-AMP degradation in the presence of OG1RF and Δ*eecP* live cells and resolved the reaction by TLC as previously done (Fig 4E). Compared to the rapid degradation of c-di-AMP, with formation of pApA, AMP, and Pi release, observed for the parent strain, the Δ*eecP* cell suspension degraded c-di-AMP slowly, with a considerable amount of c-di-AMP detected after 60 minutes (Fig 4E). As expected, the complemented Δ*eecP* strain degraded radiolabeled c-di-AMP as effectively as the parent strain, ruling out the possibility of polar effects. Collectively, these results reveal that EecP is primarily responsible for extracellular c-di-AMP degradation.

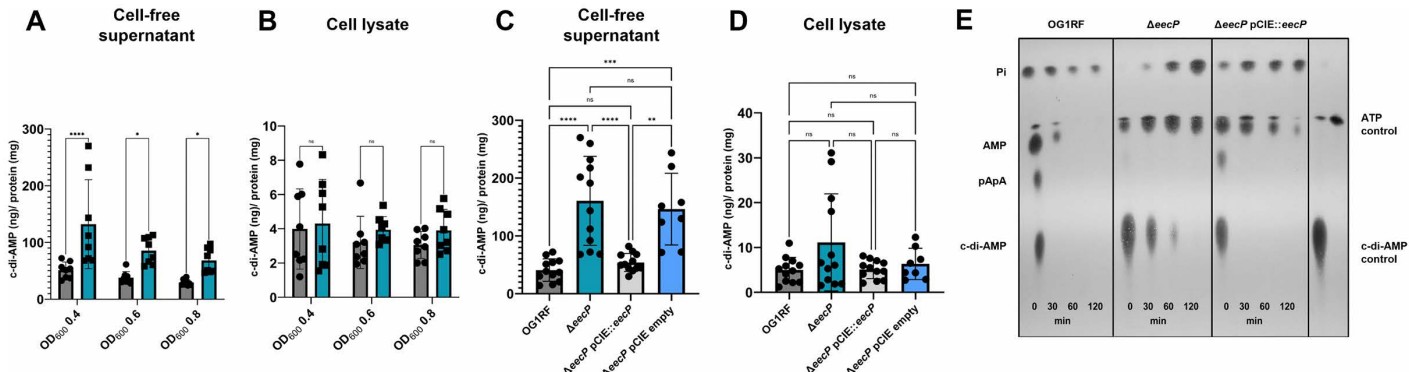

**Fig 4. Inactivation of *eecP* leads to extracellular accumulation of c-di-AMP.** The OG1RF (in grey) and Δ*eecP* (in teal) strains were grown to OD$_{600}$ of 0.4, 0.6, and 0.8, and c-di-AMP levels in cell-free supernatants (A) and cell lysates (B) were quantified using a c-di-AMP ELISA (Cayman Chemicals). **(C-D)** Complementation of Δ*eecP* (Δ*eecP* pCIE::*eecP* strain) restores c-di-AMP to parent strain levels in cell-free supernatants and cell lysates. The Δ*eecP* strain harboring an empty pCIE vector was used in the genetic complementation studies to ensure that the addition of chloramphenicol and cCF10 pheromone did not interfere with the results. ELISA data is derived from at least eight biological replicates. Statistical analysis was performed by ordinary one-way or two-way ANOVA, followed by Tukey's or Uncorrected Fisher's LSD multiple comparisons test. ****, $P \le 0.0001$, ***, $P \le 0.001$, **, $P \le 0.01$, *, $P \le 0.05$. (ns) not significant. **(E)** TLC of cell-free supernatants of OG1RF, Δ*eecP*, and Δ*eecP* pCIE::*eecP* from cultures grown in CDM to OD$_{600}$ 0.4, suspended in 50 mM Tris-Cl containing 5 mM MnCl$_2$, and spiked with [$^{32}$P]- c-di-AMP. Supernatants were sampled over time for c-di-AMP degradation. Reaction aliquots were collected at the indicated time points and inactivated by boiling before spotting on a PEI-cellulose plate for TLC separation. TLC image is a representative of experiment conducted at least two times with independent biological replicates.

## Conserved histidyl residue H203 is essential for EecP activity

To determine the individual contributions of each MP-NT domain pair to EecP activity, we introduced point mutations into the *eecP* coding sequence within the pCIE::*eecP* plasmid, originally constructed for genetic complementation of the Δ*eecP* strain. Specifically, we replaced the critical histidyl residue within the 'NHE' motif in one or both MP-NT pairs with an alanine residue to generate the SDM1 (MP-NT/D1$^{H203A}$), SDM2 (MP-NT-D2$^{H758A}$), and SDM1/2 (MP-NT/D1$^{H203A}$ and MP-NT/D2$^{H758A}$) strains (Fig 5A). These plasmids were then introduced into the Δ*eecP* background to assess the ability of the EecP variants to degrade c-di-AMP.

Using a quantitative c-di-AMP ELISA, as described above, we found that extracellular c-di-AMP levels in Δ*eecP* and SDM1 strains were comparable, indicating that the MP-NT/D1 is catalytically active (Fig 5B). On the other hand, the SDM2 strain exhibited a modest, though statistically significant, increase in c-di-AMP compared to the parent strain, which remained significantly lower than that of Δ*eecP* (Fig 5B). Notably, extracellular c-di-AMP levels were slightly higher in the SDM1/2 strain than in SDM1 alone (Fig 5B). These findings were corroborated by TLC analysis of extracellular degradation of radiolabeled c-di-AMP across the different mutants (Fig 5C). Together, these results indicate that c-di-AMP degradation is driven primarily by the first MP-NT domain pair of EecP, with the second domain pair playing a minor role.

## Cyclic di-GMP interferes with EecP c-di-AMP degradation

To obtain insights into the substrate specificity of EecP, we used the TLC-based assay to assess the capacity of unlabeled (cold) nucleotide competitors to inhibit degradation of radiolabeled c-di-AMP by *E. faecalis* OG1RF and Δ*eecP* cells. First, we performed a control experiment to determine the concentration of unlabeled c-di-AMP required to strongly inhibit degradation of [$^{32}$P]-c-di-AMP by OG1RF, showing that 100,000-X excess of cold c-di-AMP fully stabilized the [$^{32}$P]-c-di-AMP signal (S4A Fig). Further, we tested the ability of Δ*eecP* to degrade [$^{32}$P]-c-di-AMP in the presence of excess unlabeled c-di-AMP, which showed poor degradation of [$^{32}$P]-c-di-AMP in the presence or absence of the competitor (S4B Fig).

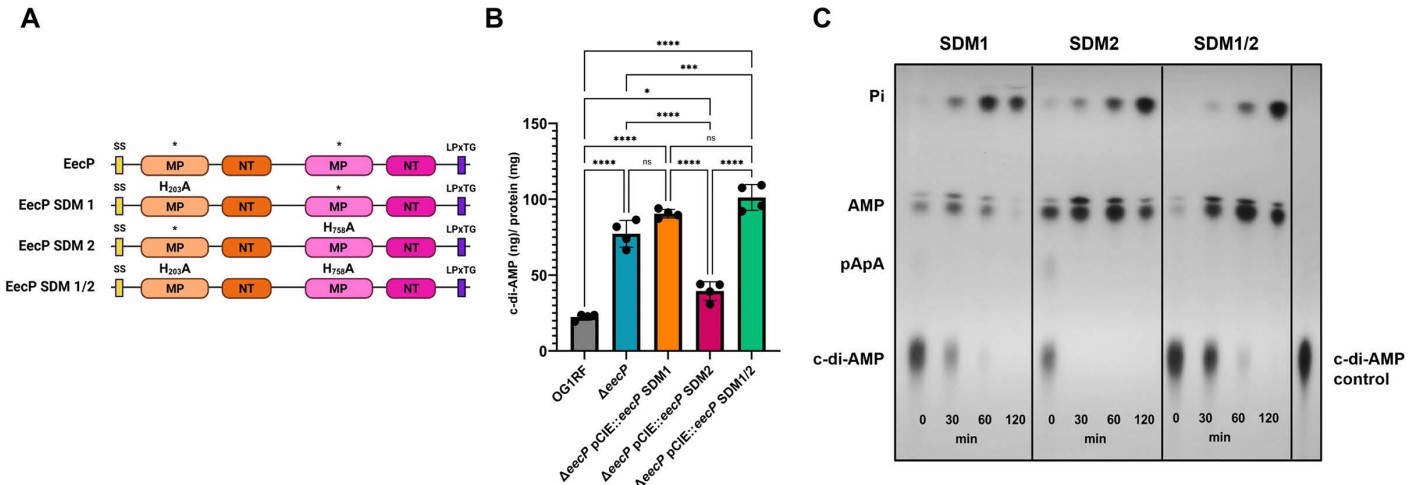

**Fig 5. The histidyl residue of the conserved 'NHE' motif of MP-NT/D1 is essential for c-di-AMP degradation. (A)** Domain structure of EecP and site-directed mutants (SDM) created. Legend: SS: secretion signal, MP: metallophosphoesterase, NT: 5′-nucleotidase, LPxTG: cell-surface anchoring motif. SDM1 has an H203A substitution in MP-NT/D1, SDM2 has an H758A substitution in MP-NT/D2, and SDM1/2 has both H203A and H758A substitutions. **(B)** Strains were grown in CDM to $OD_{600}$ 0.4, and c-di-AMP in their cell-free supernatants was quantified via ELISA. Data is derived from four biological replicates. Error bars represent the standard deviation. Statistical analysis was performed by ordinary one-way ANOVA, followed by Tukey's multiple comparisons test. ****, $P \le 0.0001$, ***, $P \le 0.001$, *, $P \le 0.05$. (ns) not significant. **(C)** TLC of cell-free supernatants of SDM1, SDM2, and SDM1/2 grown in CDM to $OD_{600}$ 0.4, suspended in 50 mM Tris-Cl containing 5 mM $MnCl_2$, and spiked with [$^{32}$P]-c-di-AMP. Supernatants were sampled over time for c-di-AMP degradation. Reaction aliquots were collected at the indicated time points and inactivated by boiling before spotting on a PEI-cellulose plate for TLC separation. TLC image is a representative of experiment conducted at least two times with independent biological replicates.

These experiments demonstrated that cold c-di-AMP could readily compete with [$^{32}$P]-c-di-AMP, significantly delaying its degradation by the parent strain (S4B Fig). Then, we tested the capacity of a panel of cold nucleotides to interfere with EecP-mediated degradation of [$^{32}$P]-c-di-AMP. We found that unlabeled c-di-GMP stabilized the [$^{32}$P]-c-di-AMP signal, almost to the same extent as c-di-AMP, suggesting that EecP might be able to broadly degrade cyclic dinucleotides (Fig 6). Consistent with the TLC profiles of OG1RF and ΔeecP strains (Fig 3), unlabeled AMP modestly delayed [$^{32}$P]-c-di-AMP degradation. In contrast, excess unlabeled cGAMP or cAMP failed to compete with [$^{32}$P]-c-di-AMP, suggesting that neither serves as a substrate for EecP (Fig 6).

### The ΔeecP strain phenocopied the parent strain under *in vitro* conditions

Having established that EecP mediates extracellular c-di-AMP degradation, we next examined whether loss of *eecP* affected growth under conditions where c-di-AMP is essential for survival or optimal proliferation. In nutrient-rich brain heart infusion (BHI) medium and chemically defined medium (CDM), the ΔeecP strain displayed growth patterns nearly identical to those of the parent strain (S5A–S5B Fig). Comparable growth was also observed in CDM supplemented with 1% peptone, 1 mM of the osmolyte glycine betaine, or 500 mM KCl: conditions previously shown to require c-di-AMP for growth [15] (S5C–S5E Fig). Since several cell wall-anchored proteins contribute to adhesion and biofilm formation, we additionally assessed the ability of the ΔeecP strain to form biofilms *in vitro*. Again, no discernible differences were observed between the parent and ΔeecP strains (S5F Fig).

### Inactivation of *eecP* impairs *E. faecalis* survival within macrophages in a cGAS-STING-dependent manner

Though we obtained compelling evidence that EecP actively degrades extracellular c-di-AMP, its role in *E. faecalis* pathophysiology remained unclear to this point. On this note, the ability of *E. faecalis* to persist and replicate in the bloodstream

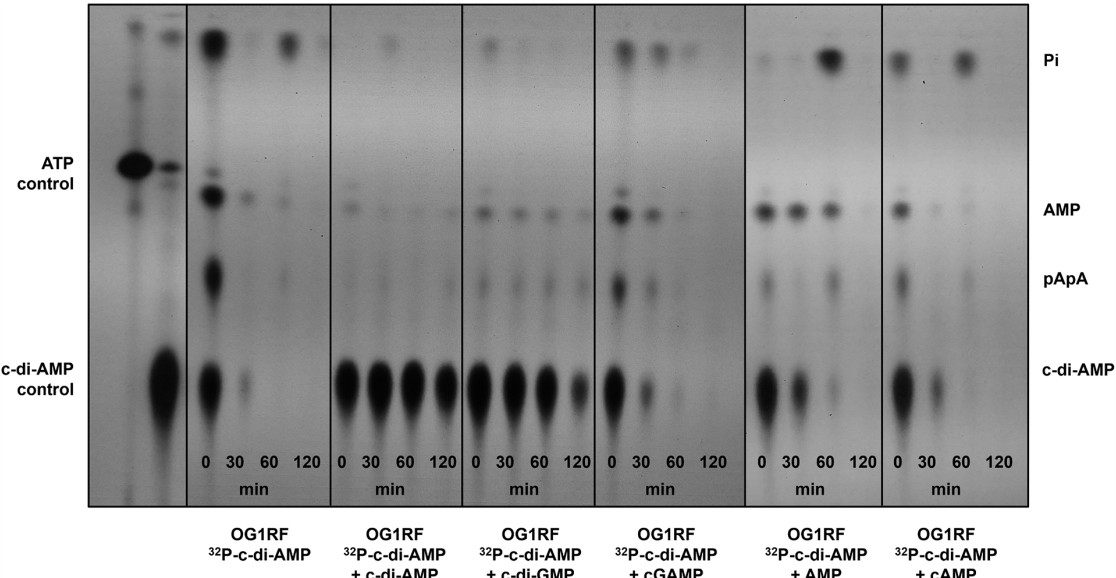

**Fig 6. Competition assays identify c-di-GMP as a potential EecP substrate.** TLC of cell-free supernatants of OG1RF cultures grown in CDM to OD$_{600}$ 0.4, suspended in 50 mM Tris-Cl containing 5 mM MnCl$_2$ and spiked with [$^{32}$P]- c-di-AMP and 100,000-X excess of cold competitor nucleotides c-di-AMP, c-di-GMP, cGAMP, AMP, or cAMP. Supernatants were sampled over time for c-di-AMP degradation. Reaction aliquots were collected at the indicated time points and inactivated by boiling before spotting on a PEI-cellulose plate for TLC separation. TLC image is a representative of experiment conducted at least two times with independent biological replicates.

and within various cell types, including neutrophils and macrophages, has been proposed as an important immune evasion strategy that is directly linked to its virulence [44–47]. Given that c-di-AMP is recognized as a pathogen-associated molecular pattern (PAMP) and a strong immunostimulant, we reasoned that the accumulation of extracellular c-di-AMP in *E. faecalis* Δ*eecP* would likely affect innate sensing and thus become relevant during host interactions. To explore this, we first tested the ability of the Δ*eecP* strain to grow or survive in human whole blood, serum, and neutrophils. Deletion of *eecP* did not impact *E. faecalis* growth in blood or serum *ex vivo* (S6A–S6B Fig). However, the Δ*eecP* mutant exhibited a trend towards reduced survival within polymorphonuclear neutrophils (PMNs) compared to the parent strain (S6C Fig).

The cGAS-STING pathway is one of the innate immune sensing mechanisms of c-di-AMP, specifically through direct activation of STING, which leads to transcriptional regulation of inflammatory cytokines, and, broadly, changes in cell survival and proliferation. Therefore, because neutrophils express low levels of STING compared to macrophages, we assessed the ability of Δ*eecP* to survive within naïve (M0) and polarized bone marrow-derived murine macrophages (BMDM) of both M1 and M2 phenotypes. Here, the Δ*eecP* strain exhibited significantly impaired survival in macrophages, regardless of their activation state (Fig 7A).

Next, we utilized inhibitors of cGAS and STING to assess the contribution of these factors to the impaired survival of Δ*eecP* within BMDMs. While treatment with the cGAS (TDI-6570) or STING inhibitors (H151) did not affect the survival of the parent strain within M1 macrophages compared to the untreated control, cGAS or STING inhibition led to significant increases in the survival rates of Δ*eecP* in M1 macrophages (Fig 7B). On the other hand, inhibition of cGAS or STING led to a modest, but statistically significant, enhancement in the survival of the parent strain, and even greater rescue of Δ*eecP* survival in M2 macrophages compared to untreated controls (Fig 7C). These results indicate that EecP contributes to *E. faecalis* survival within phagocytic cells and that the impaired survival of Δ*eecP* within murine macrophages can be attributed, at least in part, to activation of the cGAS-STING pathway.

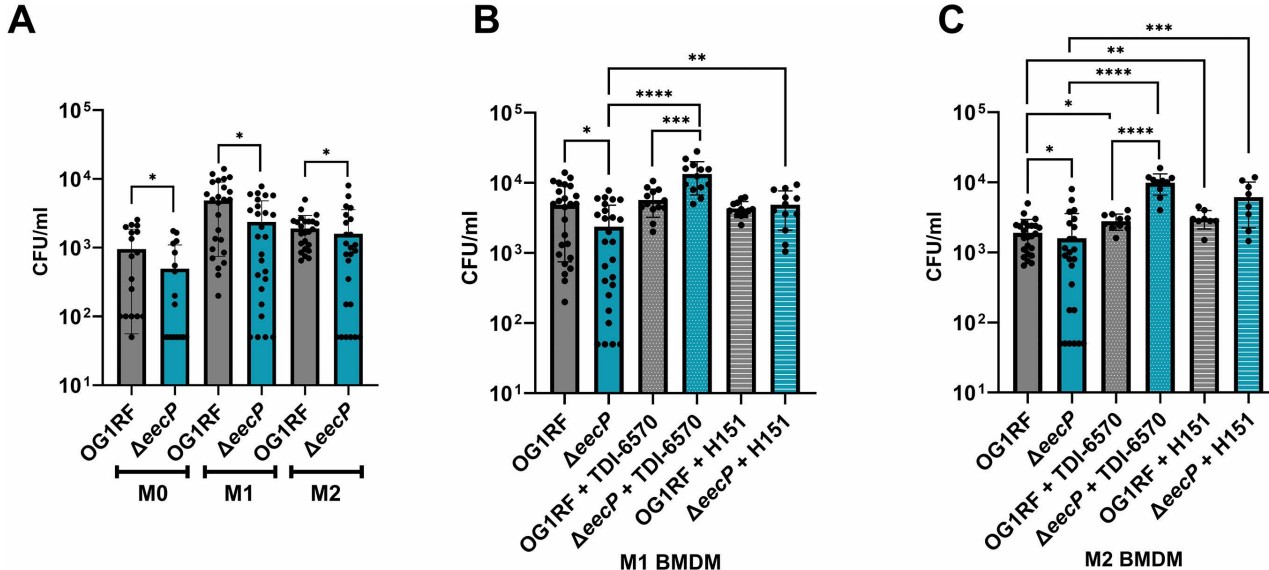

**Fig 7. ΔeecP impaired survival in differentiated murine BMDMs is restored with chemical inhibition of cGAS or STING. (A)** Survival of OG1RF and ΔeecP in undifferentiated and M1- or M2-differentiated murine BMDMs. **(B)** M1- and **(C)** M2-differentiated macrophages were treated with 1 μM of cGAS inhibitor TDI-6570 or 4 μg ml⁻¹ of STING inhibitor H151. BMDMs were infected with OG1RF (parent strain) or ΔeecP at an MOI of 1. All experiments were conducted using at least three biological replicates. Unpaired nonparametric t-test was used to determine significance, ****, P ≤ 0.0001, ***, P ≤ 0.001, **, P ≤ 0.01, *, P ≤ 0.05. Error bars represent standard deviation.

### Loss of *eecP* leads to distinct innate immune response patterns in *E. faecalis* infection of iBMDMs

To evaluate the broad impacts of extracellular c-di-AMP on host immune responses in the context of an *E. faecalis* infection, we used the NanoString nCounter Mouse Immunology Panel, which probes 561 mouse genes for innate and adaptive immune responses. Here, we obtained a snapshot of the immune gene expression profile of infected NR-9456, an immortalized mouse BMDM cell line (iBMDM). After NR-9456 infection with either OG1RF or ΔeecP strains (S7 Fig), host RNA was isolated at 0 (T0) and 2 hours (T2) after completion of the antibiotic treatment used to eliminate non-internalized bacteria. Fold changes in gene expression of iBMDMs infected with OG1RF or ΔeecP were determined by comparison to uninfected cells, where differentially expressed genes (DEGs) were identified. iBMDMs infection with OG1RF led to the significant differential expression of 79 genes immediately after completion of antibiotic protection and 166 genes 2 hours later (Fig 8A). Infection with the ΔeecP strain led to 110 DEGs after antibiotic protection and 162 DEGs after 2 hours (Fig 8A). Overall, infection with OG1RF or ΔeecP each induced a unique set of DEGs and thus distinct immune profiles (Figs 8B–8E and S8 and S1 and S2 Tables).

Among genes with the highest fold change following infection with ΔeecP, that were unchanged in OG1RF infections, were the IL-27 subunit EBI3 and the NADH oxygenase NOX4 (Fig 8B). This expression was concomitant with increased expression of the transcription factors STAT6 (Signal Transducer and Activator of Transcription 6) and ZEB1 (Zinc Finger E-Box Binding Homeobox 1) during ΔeecP infection (Fig 8B). Further, comparisons of gene expression in infected macrophages immediately after antibiotic protection (T0) revealed higher expression of 14 genes in cells infected with ΔeecP compared to OG1RF (S1 Table). Two of the most altered genes in ΔeecP-infected macrophages were NOTCH1 (Notch Receptor 1) and TBK1 (TANK Binding Kinase 1), the latter known to mediate phosphorylation of STING and IRF3 (Interferon Regulatory Factor 3) (S1 Table). However, comparison of gene expression in infected macrophages 2 hours following antibiotic protection revealed many genes expressed in lower abundance in macrophages infected with ΔeecP

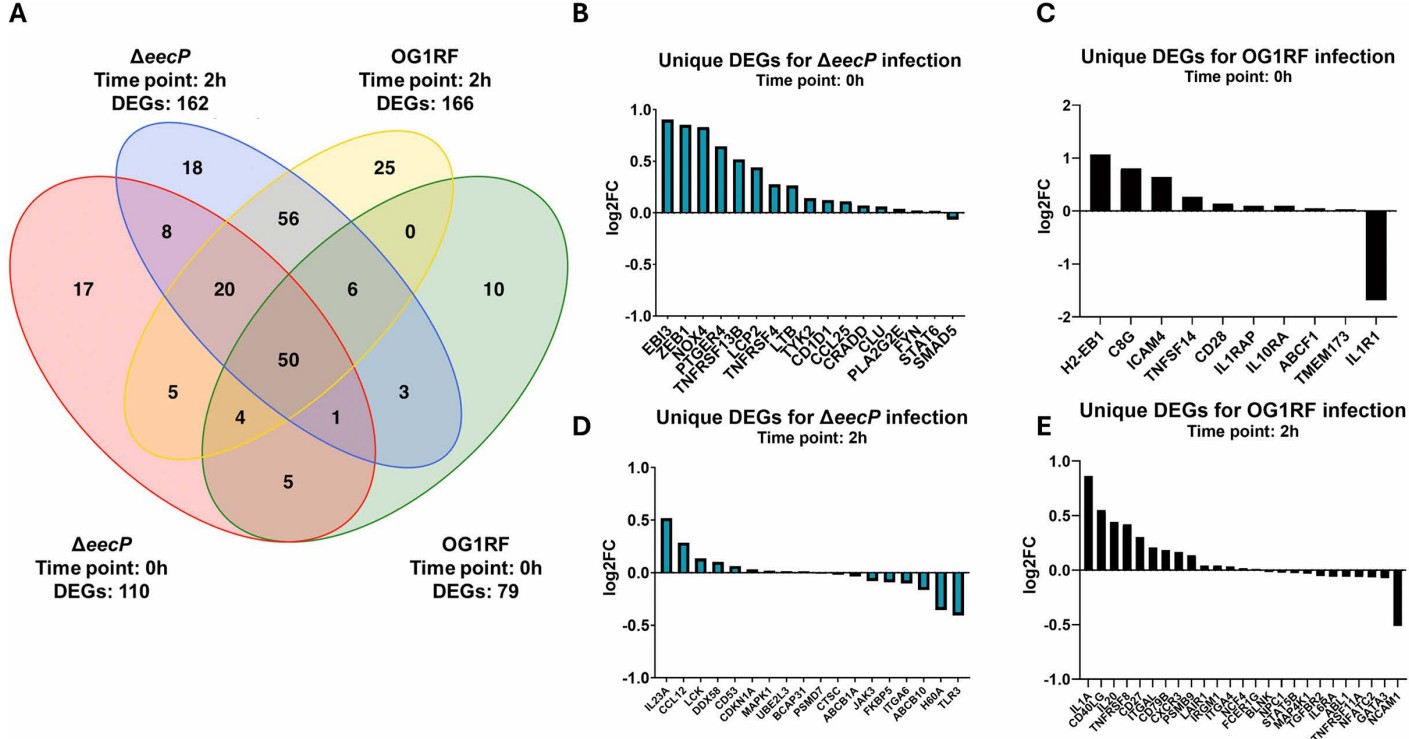

**Fig 8. Δ*eecP* infection induces distinct transcriptional responses in iBMDMs. (A)** Venn diagram showing the number of differentially expressed genes (P ≤ 0.05) in iBMDMs infected with Δ*eecP* or OG1RF at 0 h and 2 h, each compared to uninfected controls at the corresponding time point. Numbers indicate total DEGs per condition and shared genes between groups. **(B–E)** Bar plots represent significant differentially expressed genes (P ≤ 0.05) unique to each infection condition at the indicated time points, **(B)** Δ*eecP* at T0, **(C)** OG1RF at T0, **(D)** Δ*eecP* at T2, and **(E)** OG1RF at T2. Bars represent log2 fold change relative to time-matched uninfected controls.

compared to those infected with OG1RF, including MSR1 (Macrophage Scavenger Receptor 1), MARCO (Macrophage Receptor with Collagenous Structure), and IL1β (Interleukin 1 Beta) (S2 Table).

## Deletion of *eecP* impacts virulence and facilitates systemic dissemination

To directly assess the potential role of EecP in pathogenesis, we first utilized a peritoneal challenge mouse model, which leads to systemic dissemination within less than 24 hours. While no differences were observed in bacterial loads recovered from the peritoneal cavity of mice infected with either the OG1RF or Δ*eecP* strains 24 hours post-infection, a higher bacterial burden was observed in the spleens and livers of mice infected with the Δ*eecP* strain, albeit differences were only statistically significant for the liver (Fig 9A). As we suspected that the host immune responses to extracellular c-di-AMP could vary depending on the site of infection, we also utilized a catheter-associated urinary tract infection (CAUTI) mouse model [48] to evaluate the ability of Δ*eecP* to colonize the bladder, the catheter implant, and to ascend to the kidneys. At 24 hours post-infection, both OG1RF and Δ*eecP* strains were recovered in similar numbers from the implanted catheters, while mice infected with the Δ*eecP* strain exhibited a significant ~1-log reduction in bacterial burden recovered from bladders (Fig 9B). Despite this impaired bladder colonization, the Δ*eecP* strain ascended to the kidneys in ~46% of infected mice, whereas no significant bacterial colonization of the kidneys was observed in mice infected with the parental strain (Fig 9B). These findings indicate that while the impact of *eecP* deletion was niche-specific, the resulting increase in extracellular c-di-AMP may promote systemic dissemination.

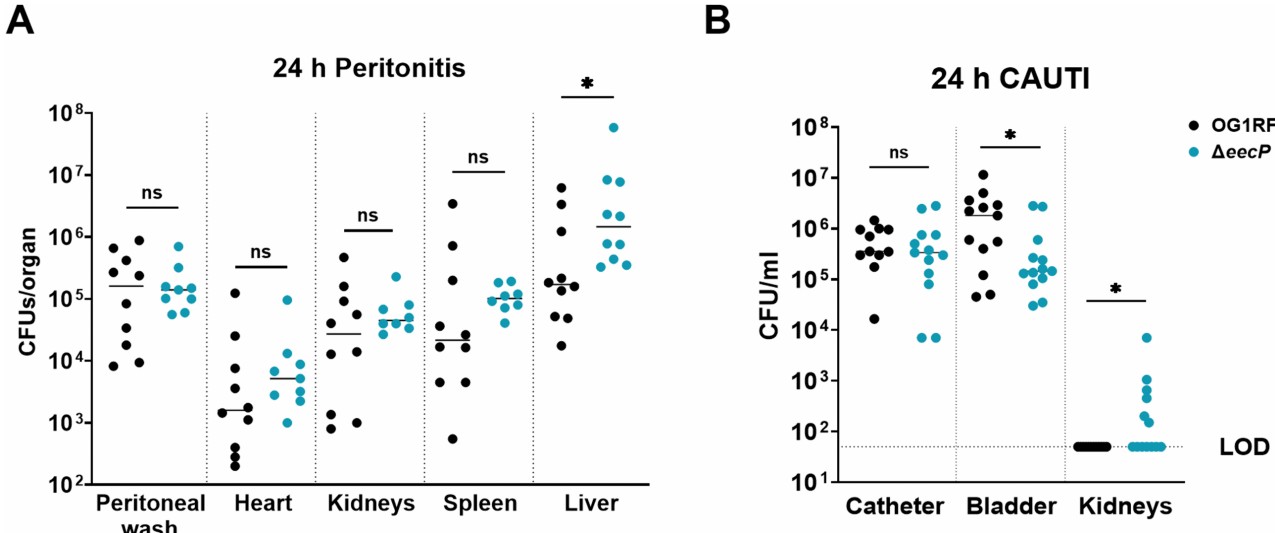

**Fig 9. Loss of *eecP* increases systemic dissemination in murine infection models. (A)** Recovery of *E. faecalis* OG1RF and Δ*eecP* from the peritoneal cavity, heart, kidneys, spleen, and liver homogenates collected from mice 24 hours post-infection. **(B)** Recovery of *E. faecalis* OG1RF and Δ*eecP* from bladder, catheter implant, and kidney homogenates collected from mice 24 hours post-infection. Data was combined from multiple independent experiments performed with at least three animals at one time. Each data point represents one animal, with each group including at least 10 animals. The data shown is the result of robust regression and outlier removal (ROUT method) of log-transformed values. Unpaired nonparametric t-test was used to determine significance, *, $P \leq 0.05$, ns, $P > 0.05$.

## Discussion

Cyclic di-AMP is a bacterial second messenger essential for the regulation of important physiological processes (e.g., osmobalance and cell wall homeostasis), stress response activation, and virulence [9–11]. In addition, c-di-AMP is recognized by host cells as a potent PAMP, alerting the host of a potential bacterial infection. In particular, c-di-AMP has been shown to bind host receptors STING [20,21,23], DDX41 [24], RECON [25], and ERAdP [26], which, in turn, activate corresponding immune signaling pathways leading to the production of multiple cytokines. While we have previously demonstrated that loss of intracellular c-di-AMP homeostasis severely impairs *E. faecalis* fitness and virulence [15], the role of extracellular c-di-AMP in *E. faecalis* interactions with the host had not been investigated. Here, we identified EecP, a previously uncharacterized ectonucleotidase, unique to *E. faecalis*, that modulates extracellular c-di-AMP levels. Using *in vitro*, *ex vivo*, and *in vivo* models, we showed that while loss of *eecP* had minimal effects on *E. faecalis in vitro* fitness, the Δ*eecP* strain was more susceptible to *ex vivo* killing by phagocytic cells and exhibited niche-specific phenotypes in mouse infection models. Given that the enzymes responsible for c-di-AMP metabolism are considered viable therapeutic targets, a deeper understanding of the roles and regulatory mechanisms governing intra- and extracellular c-di-AMP pools in bacterial pathogens such as *E. faecalis* is essential and underscores the importance of this work.

With the discovery of EecP, *E. faecalis* emerges as one of the few bacterial pathogens known to encode an extracellular c-di-AMP phosphodiesterase. However, EecP is unique in that it contains duplicated MP-NT catalytic domain pairs, distinguishing it from the ePDEs previously characterized in *S. agalactiae*, *S. suis*, and *B. anthracis*, which only contain a single MP-NT domain pair. Notably, homologs of EecP are rare among other *Enterococcus* species and absent in the human-associated *E. faecium*. TLC analysis revealed that EecP cleaves c-di-AMP into both pApA and AMP. Subsequently, pApA and AMP are degraded into Pi; therefore, we hypothesize that EecP also mediates the degradation of the c-di-AMP by-products. This hypothesis is further supported by the competition experiment showing that unlabeled AMP competes with c-di-AMP as a substrate, modestly delaying degradation of radiolabeled c-di-AMP. Previous studies on

*S. agalactiae* revealed that two ectonucleotidases with PDE activity, CdnP and NudP, act sequentially to convert c-di-AMP into AMP and subsequently into adenosine (Ado) and Pi [20,49]. Ado is a well-known immunosuppressive molecule linked to anti-inflammatory responses [50–52], and its accumulation in the extracellular space correlates with enhanced bacterial survival and virulence in both *S. agalactiae* and *S. suis* [30,49]. Given the evidence that EecP may have nucleotide substrates beyond c-di-AMP, additional experiments are required to assess the possible contribution of other nucleotides degraded by EecP to the observed immunological effects. In the absence of EecP, *E. faecalis* appears to degrade c-di-AMP much more slowly, primarily generating AMP over pApA, while both by-products rapidly disappear in the parent (EecP+) strain. Without evidence of enterococcal genomes encoding an additional surface-associated nucleotidase, *E. faecalis* may secrete an enzyme that exhibits weak activity towards c-di-AMP. It is also possible that at least a fraction of radiolabeled c-di-AMP is internalized and then degraded by intracellular PDEs. While in our experience EecP was not amenable to large-scale purification using heterologous expression systems in both *E. coli* and *B. subtilis,* as well as a cell-free expression system, our current understanding would benefit from further characterization using purified EecP. Additionally, our studies indicate that *E. faecalis* can secrete large quantities of c-di-AMP, and thus identification of its c-di-AMP export mechanisms is also warranted.

Competition experiments using unlabeled nucleotides indicated that c-di-GMP is a potential substrate of EecP degradation. Consistent with this observation, previous studies have shown that cytoplasmic c-di-AMP PDEs, as well as the CdnP (*S. agalactiae*) and SntA (*S. suis*) ePDEs, recognize c-di-GMP as a substrate [20,29,53,54]. Discovered in the late 1980s, c-di-GMP is a more comprehensively investigated bacterial second messenger shown to control numerous bacterial activities, including motility, biofilm formation and dispersion, cell differentiation, and cell division [55]. In addition, c-di-GMP functions as a signaling molecule mediating interkingdom interactions through its ability to, like c-di-AMP, serve as a PAMP molecule [56]. The identification of c-di-GMP as a potential EecP substrate suggests a possible role of this extracellular enzyme in interspecies and possibly interkingdom signaling communication. While *E. faecalis* does not possess diguanylate cyclase enzymes to synthesize c-di-GMP nor has it been shown to respond to extracellular c-di-GMP, it is possible that c-di-GMP produced and secreted by bacteria co-existing with *E. faecalis* in crowded ecosystems, such as the large intestine or polymicrobial biofilms, influences *E. faecalis* physiology. In fact, extracellular c-di-GMP has been shown to inhibit biofilm formation of other Gram-positive bacteria [57,58]. Likewise, it is possible that EecP could interfere with activities of c-di-GMP-producing organisms that may secrete and utilize c-di-GMP as an extracellular signaling molecule.

Single amino acid substitutions of conserved histidyl residues in MP domains of EecP strongly indicate that c-di-AMP hydrolysis is primarily mediated by MP-NT/D1. These findings are consistent with the observation that EecP's MP-NT/D1 is more closely related to the well-characterized catalytic domain of *S. suis* SntA than the MP-NT/D2 domain. Furthermore, a substrate-recognition motif within the NT domain of MP-NT/D1 contains the two conserved tyrosines found in CpdB-like PDEs, including *E. coli* CpdB, *S. agalactiae* CdnP, *B. anthracis* CpdB, and *S. suis* SntA (S1 Fig), that, despite some differences in substrate and cofactor specificity, have been shown to degrade c-di-AMP [20,29,31,59–62]. It follows that these tyrosine residues are replaced by two phenylalanine residues in MP-NT/D2. Of note, tyrosine and phenylalanine are aromatic amino acids that only differ by the presence of a hydroxyl group on the aromatic side chain of tyrosine. Interestingly, the phenylalanine residues of EecP MP-NT/D2 are conserved in *S. agalactiae* NudP and *E. coli* UshA, which have been shown to degrade 5′-mononucleotides [49,63] (S1 Fig). These subtle differences in key residues of NT domains in MP-NT/D1 and MP-NT/D2 hint that the duplicated MP-NT domain pairs of EecP may have different substrate specificities. At this point, we speculate that the MP-NT/D2 of EecP might be functionally related to the *S. agalactiae* NudP and capable of cleaving AMP into Ado and Pi. This possibility is supported by TLC results of site-directed mutants SDM2 and SDM1/2, which showed that while these mutants can degrade [$^{32}$P]-c-di-AMP, AMP degradation was slower compared to the SDM1 and complemented strains.

Consistent with previous findings with the *S. agalactiae* ΔcdnP strain [20], inactivation of *eecP* did not produce any noticeable *in vitro* phenotypes. To further investigate the biological significance of EecP, we assessed the ability of the

mutant to survive within phagocytic cells. Survival assays using human PMNs and murine BMDMs indicated that the ΔeecP strain is more susceptible to phagocyte killing, albeit the differences observed were only significant within BMDMs that have higher functional expression of STING compared to neutrophils [64,65]. This phenotype may be linked with activation of the cGAS-STING signaling pathway, as treatment with either cGAS or STING inhibitors alleviated the virulence attenuation of ΔeecP. Follow-up studies using cGAS and STING knock-out animals should be conducted to further probe the relationship between the cGAS-STING pathway and ΔeecP phenotypes.

NanoString analysis revealed intriguing differences in the immune signature between OG1RF and ΔeecP infections, uncovering differential expression of host targets known to be impacted by c-di-AMP, as well as novel targets. Shortly after infection, ΔeecP induced expression of EBI3, a subunit of the IL-27 cytokine, ZEB1, a transcription factor that induces genes associated with chemotaxis of immunoregulatory immune cells, and NOX4, a major producer of ROS, all known to be under transcriptional control of NFκB [66–68]. The pattern recognition receptor (PRR) and c-di-AMP receptor RECON was previously shown to negatively regulate NFκB; however, binding of c-di-AMP results in RECON inhibition followed by NFκB activation [25]. Therefore, it is not surprising that we observed increased expression of NFκB-regulated EBI3, ZEB1, and NOX4 during ΔeecP infection, a mutant with elevated c-di-AMP levels. Interestingly, these mechanisms independently and cooperatively promote a more M2-like macrophage phenotype and function [69–72].

Infection with ΔeecP also led to the increased expression of the transcription factor STAT6; its activation is dependent on the kinase TBK1 and the adaptor protein and c-di-AMP receptor STING [73]. Of note, direct comparison of ΔeecP and OG1RF infections after antibiotic protection revealed elevated expression of the kinase TBK1, a key player in the cGAS-STING pathway that mediates phosphorylation of STING and IRF3 [74,75]. Collectively, these findings agree with previous studies that propose c-di-AMP modulates STING activities and downstream responses, including NFκB activity. A unique finding of this investigation was that infection with ΔeecP led to reduced expression of macrophage receptors MSR1 and MARCO, which are integral to microbial clearance and induction of inflammation [76,77]. While this study supports an association between *E. faecalis* c-di-AMP and cGAS–STING signaling, future investigations should aim to identify if other host c-di-AMP receptors are more relevant during *E. faecalis* infections.

Although results from the two murine infection models were complex, they indicated that EecP and, by extension, extracellular accumulation of c-di-AMP play an important role in *E. faecalis* pathogenesis. The ΔeecP strain efficiently colonized the intraperitoneal cavity, which then led to increased bacterial burden in distal organs, especially the liver. Of note, *E. faecalis* has been shown to invade and divide within hepatocytes, the main epithelial cells of the liver [78]. Further, overgrowth of *E. faecalis* in the gut has been demonstrated to exacerbate ethanol-induced liver disease in mice and induce translocation of bacteria to the liver [79]. Thus, additional studies are needed to better understand the relationship between c-di-AMP sensing and *E. faecalis* colonization of the liver. In the CAUTI model, the ΔeecP strain showed a reduced ability to colonize the bladder compared to the parent strain; however, ΔeecP-infected mice had elevated infection dissemination to the kidneys. The reduction in bladder tissue colonization by the ΔeecP mutant may be due to its increased susceptibility to immune cell-mediated killing, particularly given that *E. faecalis* residing in the urothelium is more directly exposed to a robust host immune response [80]. Despite the impaired bladder colonization of ΔeecP in the CAUTI model, both murine models suggest that accumulation of extracellular c-di-AMP may positively impact systemic dissemination.

Sensing of c-di-AMP by host receptors can induce various immune signaling pathways that culminate in the production of a myriad of cytokines like IFN-β, IL-1β, IL-6, and TNFα [81]. While the induction of these cytokines is intended to limit bacterial proliferation and contain infection, their activation can be detrimental and increase host susceptibility to infections. For example, several studies have reported that type I interferons (IFNs), which are induced by c-di-AMP, enhance the susceptibility of mice to *L. monocytogenes* infection via impaired bacterial clearance, a reduction in pro-inflammatory immune cell populations, upregulation of apoptotic genes, and enhanced T cell death [82–86]. These studies serve to illustrate how excess c-di-AMP and associated activation of downstream immune responses can become detrimental to the host, enabling bacterial persistence and increasing susceptibility to infection. Along these lines, the dysregulation of

host immune responses due to sustained elevated c-di-AMP levels may explain the enhanced systemic dissemination of the Δ*eecP* strain. However, this theory warrants further investigation.

In closing, here we identified and characterized the novel ectonucleotidase EecP from the opportunistic pathogen *E. faecalis,* which emerges as an immunomodulatory factor that shapes infection outcomes. Our findings further underscore the complexity and nuanced interplay of c-di-AMP in host-pathogen interactions, highlighting the critical role of extracellular c-di-AMP as an active immunomodulator during infection.

## Materials and methods

### Ethics statement

Animal procedures for murine infections were approved by the University of Florida Institutional Animal Care and Use Committee (protocol # 202200000241) and the University of Notre Dame Institutional Animal Care and Use Committee (protocol # 22-01-6971). All animal care was consistent with the Guide for the Care and Use of Laboratory Animals from the National Research Council and the United States Department of Agriculture (USDA) Animal Care Resource Guide.

### Bacterial strains and growth conditions

The bacterial strains used in this study are listed in Table 1. *E. faecalis* strains were grown in brain heart infusion (BHI), Todd Hewitt broth (THB), or chemically defined media (CDM) at 37°C under static conditions. The CDM formulation is outlined in S3 Table. *E. coli* strains were grown in Luria broth (LB) or Super Optimal broth with Catabolite repression (SOC) at 37°C under shaking conditions. *S. agalactiae* strains were grown in THB at 37°C with 5% $CO_2$ under static conditions. Strains bearing the pCIE plasmids were grown under antibiotic pressure (10 µg ml$^{-1}$ chloramphenicol) with 5 ng ml$^{-1}$ of cCF10 pheromone (Mimotopes) added for gene activation. To determine growth kinetics, overnight cultures grown in BHI or CDM were adjusted to an optical density at 600 nm ($OD_{600}$) of ~0.2-0.3 and inoculated into fresh media at a ratio of 1:100. Growth was monitored at an $OD_{600}$ using an automated growth reader for up to 24 hours. Additional components KCl (Fisher Scientific), glycine betaine (Sigma Aldrich), and peptone (Thermo Scientific) were added to CDM at the concentrations indicated in the figure.

### General cloning techniques

The nucleotide sequences of *eecP* (OG1RF_RS00285) and *dhhP* (OG1RF_RS06025) were obtained from the *E. faecalis* OG1RF reference genome available at the NCBI website (Accession: CP002621.1). The Wizard Genomic DNA Purification kit (Promega) was used for the isolation of bacterial genomic DNA (gDNA), and the Monarch Plasmid Miniprep kit (New England Biolabs) was used for plasmid purification. The Zymo DNA Clean and Concentrator (Zymo Research) was used to isolate PCR products. Colony PCR to verify mutant identity was performed using ProMega GoTaq Green PCR 2X master mix (Promega) with primers listed in S4 Table.

### Construction of *eecP* mutant and genetically complemented strains

Deletion of *eecP* in *E. faecalis* OG1RF was carried out using the pCJK47 markerless genetic exchange system [40], as previously described. Briefly, approximately 1 kb size sequences upstream and downstream of *eecP* were amplified using the primers listed in S4 Table. The two amplicons were ligated to the pCJK47 vector, linearized with XbaI and EcoRI, and transformed into *E. coli* EC1000. Then, the vector containing the ligated amplicons was electroporated into *E. faecalis* CK111 (donor strain) that was used to deliver the plasmid to the recipient OG1RF strain, where markerless genetic exchange occurred. The gene deletion was confirmed by PCR screening the *eecP* flanking sequences, followed by Sanger sequencing and whole-genome sequencing to ensure no additional mutations have emerged. For plasmid complementation, the *eecP* gene was amplified by PCR using the primers listed in S4 Table and ligated into the pCIE

**Table 1. Strains used in this study.**

| Strain | Relevant characteristics | Source |
|---|---|---|
| *E. faecalis* | | |
| OG1RF | Laboratory/ reference strain Rif$^r$, Fus$^r$ | Lab stock |
| CK111 | OG1Sp *upp4*::P$_{23}$*repA4*, Spec$^r$ | Lab stock |
| V583 | Laboratory/ reference strain, Van$^r$ | Lab stock |
| OG1RF Δ*eecP* | *eecP* (OG1RF_10056, RS00285) | This study |
| OG1RF pCIE::empty | pCIE empty vector, ccF10 pheromone inducible, Cm$^r$ | Lab stock |
| OG1RF Δ*eecP* pCIE::*eecP* | ccF10 pheromone inducible, Cm$^r$ | This study |
| OG1RF Δ*eecP* pCIE::*eecP* SDM1 | ccF10 pheromone inducible, Cm$^r$ | This study |
| OG1RF Δ*eecP* pCIE::*eecP* SDM2 | ccF10 pheromone inducible, Cm$^r$ | This study |
| OG1RF Δ*eecP* pCIE::*eecP* SDM 1/2 | ccF10 pheromone inducible, Cm$^r$ | This study |
| *E. coli* | | |
| EC1000 | Host for cloning RepA-dependent plasmids | Lab stock |
| DH10β | Host for cloning | Lab stock |
| NEB 5-alpha | T1 phage resistant and *endA* deficient | New England BioLabs |
| BL21 (DE3) pLyss pET30::*dhhP* | Host for rDhhP protein expression, Kn$^r$ | This study |
| Rosetta (DE3) pLysS pSpeedET::*Akr1c13* | Host for AKR1C13, protein herein referred to as RECON, protein expression, Kn$^r$ | Woodward Lab at the University of Washington |
| Rosetta (DE3) pLysS pET20b::*disA* | Host for DisA protein expression (*disA* from *Bacillus subtilis*), Amp$^r$ | Woodward Lab at the University of Washington |
| *S. agalactiae* | | |
| COH1 | Laboratory/ reference strain, serotype III | Brady Lab at the University of Florida |
| NGBS93 | WGS clinical blood isolate, serotype V, BioSample: SAMN03329865 | Fittipaldi Lab at the University of Montreal |

vector linearized with BamHI and HindIII restriction enzymes to yield the plasmid pCIE::*eecP*. The plasmid was propagated in *E. coli* EC1000, verified by sequencing, and electroporated into the *E. faecalis* Δ*eecP* strain as described elsewhere [40].

## Construction of site-directed mutant complemented strains

The vector pCIE::*eecP* was isolated from the propagating *E. coli* EC1000 strain. Then, using the Q5 Site-Directed Mutagenesis Kit (New England Biolabs), a histidine to alanine substitution was performed following the manufacturer's instructions. Briefly, the substitution was created by incorporating the nucleotide change in the center of the forward primer, including at least 10 complementary nucleotides on the 3′ side of the mutation, and ensuring the 5′ ends of the two primers annealed back-to-back when designing the reverse primer. Following PCR amplification of the plasmid containing the substitution, the reaction was incubated with an enzyme mix containing a kinase, a ligase, and DpnI. These enzymes ensure the circularization of the PCR product and the removal of the template DNA. The resulting plasmid was transformed into 5-alpha competent *E. coli* (New England Biolabs) and, after isolation of the plasmid from the propagating strain, electroporated into *E. faecalis* Δ*eecP*. The mutagenized plasmids were confirmed by whole plasmid sequencing.

## Construction of recombinant DhhP expression strain

The *dhhP* gene was amplified by PCR using the primers listed in S4 Table and ligated into the pET30 vector linearized with BamHI and XhoI restriction enzymes to yield the plasmid pET30::*dhhP*. The plasmid was propagated in *E. coli* DH10β, verified by sequencing, and transformed into *E. coli* BL21 (DE3) pLyss.

## Recombinant protein expression and purification

Recombinant 6xHis-tagged *E. faecalis* DhhP, *B. subtilis* DisA, and mRECON proteins were expressed and purified from strains listed in Table 1. Briefly, bacterial cultures transformed with expression plasmids were grown in LB containing 0.04% (w/v) glucose and 300 µg ml$^{-1}$ kanamycin or 100 µg ml$^{-1}$ ampicillin overnight, sub-cultured to an $OD_{600}$ of 0.05, and allowed to reach $OD_{600}$ 0.5 before induction. Protein expression was induced using the allolactose homolog isopropyl ß-D-1-thiogalactopyranoside (IPTG) at a concentration of 500 µM for 3 hours at 30°C. Induction was confirmed by SDS-PAGE, followed by Coomassie blue staining. After expression, cell pellets were collected by centrifugation and stored at -20°C. Pellets were resuspended in equilibration buffer (20 mM $NaH_2PO_4$, 300 mM NaCl, 10 mM imidazole, pH 7.5) and added protease inhibitor cocktail (Halt Protease Inhibitor Cocktail from Thermo Scientific). Protein lysates were obtained by sonication using a 10-minute cycle of alternating pulses at 10% amplitude. Soluble protein fractions were collected by centrifugation and used to purify the recombinant protein by affinity chromatography using a nickel-nitrilotriacetic acid (Ni-NTA) matrix. Briefly, clear lysate was combined with Ni-NTA agarose (Marvelgent Biosciences) and allowed to rotate at 4°C for 2 hours. The agarose-lysate mix was then added to a crystal column and allowed to settle by gravity. After collecting flow-through lysate, the column was washed thrice with three different buffer preparations: wash buffer 1: 20 mM NaH$_2$PO4, 300 mM NaCl, 25 mM imidazole, 1% Triton x-100, wash buffer 2: 20 mM NaH$_2$PO4, 500 mM NaCl, 25 mM imidazole, and wash buffer 3: 20 mM NaH$_2$PO4, 300 mM NaCl, 50 mM imidazole. Protein was eluted four times with 200 mM $NaH_2PO_4$, 300 mM NaCl, and 250 mM imidazole. Protein elution fractions were concentrated using the Pierce Protein Concentrator PES 10K MWCO (5–20 ml) (Thermo Scientific), followed by desalting/ buffer exchange using the Zeba spin columns 7K MWCO (Thermo Scientific). The buffer used for the exchange for DhhP was 50 mM Tris-Cl, 5 mM $MnCl_2$. Concentrated DisA and mRECON were buffer exchanged with DisA activity assay buffer containing 40 mM Tris, pH 7.5, 100 mM NaCl, 20 mM $MgCl_2$. A final concentration step was performed using the Pierce Protein Concentrator PES 10K MWCO (100–500 µl) (Thermo Scientific).

## Protein structure modeling

Domain mapping and annotation were performed with Pfam, data now available through InterPro. AlphaFold Server was used to generate all predicted structures. PyMol was used for structure alignment and highlighting of structures and residues. RMSD and TM scores were determined using the Protein Data Bank (PDB) Pairwise Structure Alignment tool.

## c-di-AMP quantifications by ELISA

Overnight cultures in CDM were diluted in fresh media and allowed to reach the desired $OD_{600}$. To quantify extracellular c-di-AMP, cultures were pelleted by centrifugation, and supernatant was collected. Supernatants were centrifuged again and then filtered using 0.2 µm filters (Cytiva). For intracellular c-di-AMP quantification, pellets were collected by centrifugation, washed thrice with phosphate-buffered saline (PBS), and resuspended in 50 mM Tris-HCl, pH 7.5. Cell suspensions were transferred to screwcap tubes containing glass beads and lysed by bead beating, followed by heat inactivation at 95°C for 10 min. Then, samples were centrifuged, and supernatants were collected for intracellular c-di-AMP quantification. A commercial ELISA (Cayman Chemicals) was used for c-di-AMP quantification following the manufacturer's instructions. Intra- and extracellular samples were diluted as needed to interpolate c-di-AMP concentration in the standard curve.

## Synthesis of [α-$^{32}$P] labeled c-di-AMP

Radiolabeled c-di-AMP was synthesized as follows: 1 µM [α-$^{32}$P]-ATP (Revvity) was incubated with 1 µM of the diadenylate cyclase protein DisA in 40 mM Tris pH 7.5, 100 mM NaCl, 20 mM MgCl$_2$ binding buffer at 37 °C overnight. The mixture was boiled for 5 min, followed by centrifugation to remove precipitated DisA. Synthesized [$^{32}$P]-c-di-AMP was further purified from the mixture using recombinant His-tagged mRECON, a c-di-AMP binding protein, by affinity purification. Then, 50 µM mRECON was bound to Ni-NTA resin (Marvelgent Biosciences) for 30 min on ice with agitation. The resin was washed twice with the binding buffer to remove unbound mRECON. The resulting resin was incubated with the crude [$^{32}$P]-c-di-AMP for 30 min at room temperature, followed by 10 minutes on ice. After removing the supernatant, the Ni-NTA resin was washed twice with ice-cold binding buffer and incubated with 250 µl of binding buffer for 10 minutes at 95 °C. The slurry was then transferred to a mini-spin column (Thermo Scientific) and centrifuged for 1 min to elute [$^{32}$P]-c-di-AMP. The purity of the final preparation was evaluated by thin-layer chromatography (TLC). For TLC, polyethylenimine (PEI) cellulose F (EMD Millipore) plates were soaked in 90% methanol and then air dried. 1 µl of final preparation was resolved on TLC plates in buffer containing 1:1.5 (v/v) saturated (NH$_4$)$_2$SO$_4$ and 1.5 M NaH$_2$PO$_4$, pH 3.6. Final yield was determined by measuring the radioactivity counts of the diluted preparation in a liquid scintillation counter. The yield was ~3.3 nM [$^{32}$P]-c-di-AMP on average.

## c-di-AMP degradation assay and thin-layer chromatography

Overnight cultures grown in CDM or THB were diluted in fresh media and allowed to reach mid-log (~OD$_{600}$ 0.4). Pellets were collected by centrifugation, washed thrice with 50 mM Tris-HCl, pH 7.5, 5 mM MnCl$_2$, and resuspended to a cell density ranging from $10^8$ to $10^9$ CFU ml$^{-1}$. To the cell suspension, [$^{32}$P]-c-di-AMP was added in a 1:10 volume ratio (1 µl c-di-AMP per 10 µl cells). Cell suspensions were incubated at 37°C, aliquots collected at selected time points, and immediately centrifuged to collect supernatants, followed by heat inactivation for 5 minutes at 95°C. For competition assays, an excess of cold nucleotides (c-di-AMP, 5′-AMP, cAMP, c-di-GMP, and 2′3′-cGAMP) was added to the initial cell suspension along with radiolabeled c-di-AMP. Samples were processed as indicated before, and 3–5 µl of reaction products were spotted and resolved on PEI cellulose TLC plates. Plates were air-dried and exposed to X-ray films for an appropriate time, followed by visualization using an X-ray film processor.

## Biofilm assay

Overnight cultures grown in BHI were pelleted by centrifugation and washed thrice with PBS. Cultures were then normalized to an ~OD$_{600}$ of 0.5 and inoculated into BHI at a 1:100 ratio in 96-well polystyrene plates. Inoculated plates were incubated at 37°C under static conditions for 24 hours. Following incubation, planktonic cultures were removed from the plates, wells were gently washed with PBS twice, and plates were blot dried. Then, the adherent cells were stained with 0.1% crystal violet for 20 min at room temperature. After incubation, the dye was removed from the wells, plates were washed twice with PBS, followed by blot drying. Finally, the bound dye was eluted in a 33% acetic acid solution for 15 min before reading absorbance at 595 nm using a Synergy H1 microplate reader (Biotek).

## Growth and survival in human blood and serum

Blood was purchased from LifeSouth Community Blood Center (Gainesville, FL). This procedure was approved and performed in compliance with the University of Florida Institutional Review Board (IRB) Study #IRB 202100899. The experiments were performed with pooled blood and isolated serum from two donors of the same blood type. For isolation of pooled sera, pooled whole blood was left to stand at room temperature for 30 minutes and centrifuged at 2000 rpm for 10 minutes. The resulting serum supernatant was aliquoted and stored at −20°C for future use. For both blood and serum infections, overnight cultures grown in BHI were pelleted by centrifugation, washed thrice in PBS, normalized to an ~OD$_{600}$

of 0.5, and inoculated at a 1:1,000 ratio into blood or serum. Samples were incubated at 37°C with constant rotation. At specific time points, samples were serially diluted in PBS and plated on BHI agar to determine colony-forming units (CFU).

## Isolation of neutrophils (PMNs)

Human neutrophils were isolated as previously described [87]. Buffy coats from 6 to 8 donors were purchased from Life-South Community Blood Center and processed on the day of the experiment. Briefly, an equal volume of dextran-heparin buffer (3% (w/v) Dextran-500, 0.00568% (w/v) of heparin, 0.9% NaCl) was added to each donor's buffy coat and incubated at 37°C for 1 hour. After incubation, the high leukocyte and low erythrocyte supernatant was transferred to a new tube. Then, the leukocyte pellet of each donor was collected by centrifugation at 500 g for 10 minutes at 4°C and pooled together in 30 ml of dextran-heparin buffer. The pooled leukocyte suspension was underlaid with 10 ml of Histopaque 1077 (density, 1.077 g ml$^{-1}$) and centrifuged at 400 g for 40 minutes at room temperature to isolate the granulocytes and any remaining erythrocytes (PMN-erythrocyte layer). The supernatant above this layer was discarded, and the PMN-erythrocyte cell pellet was resuspended in 1% (w/v) ammonium chloride for 2–3 min for hypotonic lysis of remaining erythrocytes. The complete lysis of erythrocytes and neutrophil isolation was achieved by repeating the hypotonic lysis step and centrifugation at 400 g for 5 minutes at 20°C. Trypan blue staining was used to determine the number of live neutrophils.

## PMNs infection survival assay

Approximately 1 x 10$^6$ freshly isolated human polymorphonuclear cells (PMNs) were resuspended in Roswell Park Memorial Institute (RPMI) 1640 medium (Thermo Scientific) supplemented with 15% (v/v) inactivated fetal bovine serum (FBS) (Gibco) and added to a 24-well flat-bottom tissue-culture treated plate. PMNs were infected with ~1 x 10$^7$ *E. faecalis* cells for a multiplicity of infection (MOI) of 1:10 neutrophil to bacteria ratio. After the addition of bacteria, plates were incubated in a shaking incubator at 37°C with agitation at 50 rpm for 90 min. The percentage of bacterial survival was calculated by enumerating the CFU of bacteria in the wells that had neutrophils compared to bacteria-only controls.

## Isolation and polarization of bone-marrow-derived monocytes (BMDMs)

Bone marrow from femurs and tibias of 6- to 8-week-old female C57BL/6 mice was flushed with RPMI 1640 media supplemented with 1X L-glutamine, 5.95 g L$^{-1}$ HEPES, and 1 mM sodium pyruvate (VWR) and filtered using a 40 μm pore size cell strainer (VWR). Peripheral blood mononuclear cells (PBMCs) were separated from red blood cells and neutrophils using centrifugation separation with density gradients Histopaque-1077 (Sigma-Aldrich). After centrifugation, the PBMCs layer was isolated and washed with PBS. Then, PBMCs were resuspended in Dulbecco's Modified Eagle Medium (DMEM) supplemented with 4.5 g L$^{-1}$ glucose, 2 mM L-glutamine, 1 mM sodium pyruvate (VWR), 10% FBS, 100 units ml$^{-1}$ penicillin, and 100 μg ml$^{-1}$ streptomycin and incubated in cell culture flasks (Greiner Bio-One) for 24 hours at 37°C in a 5% CO$_2$ incubator. After 24 hours, 25 ng ml$^{-1}$ recombinant mouse M-CSF (R&D Systems) was added to the culture for 5–7 days for differentiation into macrophages. Uninduced (M0) BMDMs were cultured in their respective media without cytokine treatment. For M1 polarization, cells were treated for 24 hours with 100 ng ml$^{-1}$ GM-CSF (Shenandoah Biotechnology), and with 100 ng ml$^{-1}$ IL-4 (PeproTech) for M2 polarization.

## BMDMs antibiotic protection assay

Approximately 2 x 10$^4$ BMDMs were seeded per well in a 96-well flat-bottom tissue-culture treated plate (Greiner Bio-One). Macrophages were stimulated for polarization with the listed concentrations as described above. During the 24-hour stimulation, macrophages were cultured in DMEM supplemented with 10% FBS, 100 units ml$^{-1}$ penicillin, and 100 μg ml$^{-1}$ streptomycin and incubated at 37°C in 5% CO$_2$. After 24 hours, the media was removed, the cells washed thrice with PBS and replaced with fresh media free of antibiotics. Cells were then infected with *E. faecalis* strains at an MOI ratio of 1 and

incubated for 45 minutes at 37°C in a 5% $CO_2$ incubator to allow bacteria uptake. After 45 minutes, supernatants were transferred into a 96-well round-bottom plate for serial dilution and CFU determination. Macrophages were washed thrice with PBS before being replaced with DMEM supplemented with 200 µg ml$^{-1}$ gentamicin and 1X penicillin/streptomycin. After incubation at 37°C, 5% $CO_2$ for 3 h, the media was removed, and the cells were washed three times with PBS before treatment with distilled water for 15 minutes to lyse macrophages. Water was then transferred into a 96-well plate for dilutions, plated on selective BHI media containing rifampicin and fusidic acid, and incubated overnight at 37°C for CFU determination.

## BMDMs antibiotic protection assay with cGAS and STING inhibitors

BMDMs were isolated and polarized as indicated above. In addition to the cytokines (GM-CSF or IL-4), inhibitors H-151 (4 µg ml$^{-1}$) or TDI-6570 (1 µM) were added to each well concurrently. After 24 hours, the media was removed, the cells washed thrice with PBS and replaced with fresh media free of antibiotics with the addition of H-151 (4 µg ml$^{-1}$) or TDI-6570 (1 µM). Cells were then infected with *E. faecalis* strains at an MOI ratio of 1 and incubated for 45 minutes at 37°C in a 5% $CO_2$ incubator to allow bacteria uptake. After 45 minutes, supernatants were transferred into a 96-well round-bottom plate for serial dilution and CFU determination. Macrophages were washed thrice with PBS before being replaced with DMEM supplemented with 200 µg ml$^{-1}$ gentamicin and 1X penicillin/streptomycin and H-151 (4 µg ml$^{-1}$) or TDI-6570 (1 µM). After incubation for 3 h at 37°C and 5% CO2, cells were processed as described above for CFU determination.

## NanoString analysis of infected iBMDMs

The murine macrophage cell line, NR-9456 (BEI Resources), was derived using primary bone marrow cells from wild-type mice, which were immortalized by transducing them with a replication-deficient retrovirus (J2). iBMDMs were maintained in DMEM high glucose, pyruvate supplemented with 10% FBS and Penicillin-Streptomycin-Glutamine 1X, all Gibco reagents obtained from Thermo Scientific, in a humidified incubator at 37°C and 5% $CO_2$. For infections, cells were transferred to 24-well plates and allowed to attach overnight (15–18 hours). Before infections, maintenance media was removed, cells washed gently with PBS and replaced with complete media without antibiotics. Cells were infected at an MOI of 20–50 and incubated for 1 hour at 37°C and 5% $CO_2$ to allow for bacterial attachment and phagocytosis. Antibiotic protection was performed for 1 hour using 300 µg ml$^{-1}$ gentamycin, 50 µg ml$^{-1}$ penicillin, and 100 µg ml$^{-1}$ vancomycin to eliminate extracellular bacteria. At determined time points, the media was removed, replaced with 1 volume of 0.01% Triton X-100, and incubated for 5 min at room temperature for immune cell lysis and bacterial recovery. CFU was determined by serial dilution of immune cell lysates in PBS and plating in BHI agar.

For RNA isolation, at determined time points, the media was removed, and cells were resuspended in 350 µl of lysis buffer containing β-mercaptoethanol from the Qiagen RNA Mini Kit (Qiagen, USA). Samples were stored at -80°C for subsequent RNA extraction. RNA isolation and DNase digestion were performed following the manufacturer's guidelines (Qiagen, USA). NanoString hybridization was performed on the nCounter system using the NanoString murine immunology panel, which probes 561 mouse genes for innate and adaptive immune responses, as previously described [88]. This multiplex technique is an amplification-free technology that measures gene expression by counting mRNA molecules directly with barcoding to allow multiplex analysis of multiple genes in the same sample. Briefly, a total of 20 ng mRNA from each sample was hybridized to reporter-captured probe pairs at 65 °C for 18 hours. After this solution-phase hybridization, the nCounter prep station was used to remove excess probes, align the probe/target complexes, and immobilize these complexes in the nCounter cartridge. The nCounter cartridge was then placed in a digital analyzer for image acquisition and data processing. Fluorescent color codes designating mRNA targets of interest were directly imaged on the surface of the cartridge. The expression level of each gene was measured by counting the number of times the color-coded barcode for that gene was detected, and the barcode counts were tabulated. Background thresholding and normalization were performed using the Rosalind software analysis program. Differential gene expression was generated using

logarithmically transformed fold changes of averaged normalized counts for each cell population. Normalized NanoString counts were analyzed using the R statistical environment [89] (https://www.r-project.org/). Heatmaps were generated with the pheatmap package [90], and Venn Diagram analysis was conducted with the package VennDiagram [91]. Differential expression between experimental groups and the reference control was assessed using two-tailed Student's t-tests.

### Intraperitoneal challenge murine model

Murine peritonitis experiments were performed under protocol #202200000241, approved by the University of Florida Institutional Animal Care and Use Committee (IACUC). The murine peritonitis infection model has been described elsewhere [92]. Briefly, bacterial strains for inoculations were grown in BHI to an $OD_{600}$ 0.5. Pellets were collected by centrifugation, washed thrice in PBS, and suspended at ~$2 \times 10^8$ CFU $ml^{-1}$. Seven-week-old C57BL6/J mice purchased from Jackson Laboratory were intraperitoneally injected with 1 ml of bacterial suspension and euthanized by $CO_2$ asphyxiation 24 hours post-infection. To collect the peritoneal wash, the abdomen was opened to expose the peritoneal lining, and 5 ml of cold PBS were injected into the peritoneal cavity, with 4 ml retrieved as the peritoneal wash. Following peritoneal wash collection, the spleen, kidneys, liver, and heart were surgically removed, briefly rinsed in 70% ethanol, and then in sterile PBS. Spleens, kidneys, and hearts were homogenized in 1 ml of PBS, and livers in 2 ml of PBS. All sample homogenates were serially diluted in PBS and plated on selective BHI plates containing rifampicin and fusidic acid.

### Murine catheter-associated urinary tract infection model

Mice were subjected to transurethral implantation of a silicone catheter and inoculated as previously described [48]. Briefly, ~6-week-old female C57BL/6J mice purchased from Jackson Laboratory were anesthetized by isoflurane inhalation and implanted with a 6-mm-long silicone catheter (Braintree Scientific). Immediately following catheter implantation, animals were infected with 50 µl of ~$1 \times 10^6$ CFU $ml^{-1}$ of bacterial suspension in PBS. Bacteria were introduced into the bladder lumen by transurethral inoculation. Mice were euthanized 24 hours post-catheterization and infection by anesthesia inhalation, followed by cervical dislocation. Catheters were collected, and the bladder and kidneys were aseptically harvested. Kidneys were homogenized, and catheters were cut into small pieces and sonicated for CFU enumerations. All homogenates were plated on selective BHI plates containing fusidic acid and rifampicin.

### Supporting information

**S1 Data. Data.**
(ZIP)

**S1 Fig. Conservation of key histidyl and tyrosine residues among representative nucleotidases with MP-NT domains.** Alignment was performed with Multiple Sequence Comparison by Log-Expectation (MUSCLE) via SnapGene. Sequence sections represented contain conserved histidyl (H) residues found in metallophosphoesterase (MP) domains and tyrosine (Y) or phenylalanine (F) residues found in nucleotidase (NT) domains, all of which are highlighted in blue. Additional conserved residues are highlighted in yellow. Color blocks at the top represent conservation scores for a given residue, with dark red indicating the highest conservation and dark blue indicating the lowest, as calculated by SnapGene. Protein sequences were obtained from genomes of representative strains *B. anthracis* (Sterne), *E. coli* (K12 MG1655), *S. agalactiae* (NEM316), *S. suis* (SC19), and *E. faecalis* (OG1RF) available in the NCBI or BioCyc databases.
(TIF)

**S2 Fig. Recombinant *E. faecalis* DhhP (rDhhP) degrades c-di-AMP.** (A) 50 µM of purified rDhhP was mixed with radiolabeled [$^{32}$P]-c-di-AMP in a 50 mM Tris-HCl, 5 mM $MnCl_2$ buffer. Reaction aliquots were collected at the indicated time points and inactivated by boiling before being resolved via TLC. (B) [$^{32}$P]- c-di-AMP suspended in reaction buffer was incubated at 37°C for 30 min before inactivation by boiling. The inactivated suspension was then incubated at room

temperature and spotted to check for spontaneous degradation for up to 6 h. TLC image is a representative of experiment conducted at least two times.
(TIF)

**S3 Fig. EecP is a cell wall-associated protein.** OG1RF and Δ*eecP* strains were grown to an ∼OD$_{600}$ 0.8, and cell pellets treated with lysozyme (10 mg ml$^{-1}$) followed by ultracentrifugation to isolate cell wall fractions. Samples were concentrated 100-fold with a 100 kD cut-off Amicon filter and separated by SDS-PAGE. Gels were stained with SYPRO Ruby and visualized with a UV source. The red box highlights the EecP band in OG1RF lanes, which was confirmed by mass spectrometry. Gel image is a representative of experiment conducted at least two times with independent biological replicates.
(TIF)

**S4 Fig. Extracellular [$^{32}$P]- c-di-AMP degradation can be inhibited with excess cold c-di-AMP.** TLC of cell-free supernatants of OG1RF (A-B) or Δ*eecP* (B) from cultures grown in CDM to ~OD$_{600}$ 0.4, suspended in 50 mM Tris-Cl containing 5 mM MnCl$_2$, and spiked with [$^{32}$P]- c-di-AMP and cold c-di-AMP (competitor) in 10,000 or 100,000-X excess. Suspensions were sampled over time for c-di-AMP degradation. Reaction aliquots were collected at the indicated time points and inactivated by boiling before spotting on a PEI-cellulose plate for TLC separation. TLC image is a representative of experiment conducted at least two times with independent biological replicates.
(TIF)

**S5 Fig. Deletion of *eecP* does not impact *E. faecalis in vitro* phenotypes.** Growth curves of OG1RF and Δ*eecP* in (A) brain heart infusion (BHI), (B) chemically defined media (CDM), or CDM supplemented with (C) 1% peptone, (D) 1 mM glycine betaine, or (E) 500 mM KCl. Curves represent the average derived from at least three biological replicates. Error bars represent standard deviation. (F) Biofilm biomass quantification of parent strain OG1RF and Δ*eecP* grown in 96-well plates in BHI for 24 hours. Data points represent fifteen biological replicates from five independent experiments. Unpaired nonparametric t-test was used to determine significance, ****, $P \leq 0.0001$, ***, $P \leq 0.001$, **, $P \leq 0.01$, *, $P \leq 0.05$, ns, $P \geq 0.05$. Error bars represent the standard error margin.
(TIF)

**S6 Fig. Inactivation of *eecP* differentially impacts growth and viability of *E. faecalis ex vivo*.** Growth of the parent strain OG1RF and Δ*eecP* in (A) human blood and (B) human serum. Viability of OG1RF and Δ*eecP* in (C) polymorphonuclear cells (PMNs). All experiments were conducted using at least three biological replicates. Unpaired nonparametric t-test was used to determine significance, ****, $P \leq 0.0001$, ***, $P \leq 0.001$, **, $P \leq 0.01$, *, $P \leq 0.05$, ns, $P \geq 0.05$. Error bars represent standard deviation.
(TIF)

**S7 Fig. Loss of *eecP* does not impact bacterial or macrophage viability in immortalized BMDM infection.** A. Viability of OG1RF and Δ*eecP* in iBMDMs. B. Viability of iBMDMs 6 h post-infection and antibiotic protection. Macrophage viability was determined with Trypan blue staining and enumerated with a Countess II automated cell counter. Experiments were conducted using at least two biological replicates. Error bars represent standard deviation.
(TIF)

**S8 Fig. Loss of *eecP* is associated with a distinct innate immune profile in infected iBMDMs.** Heatmaps show analytes with significant differences across groups (one-way ANOVA, $P \leq 0.05$) at 0 h (A) and 2 h (B). Counts were averaged across biological replicates (n = 4 per group) for visualization and displayed as standardized values (mean = 0, SD = 1). Hierarchical clustering was performed on both rows and columns independently for each time point.
(TIF)

**S1 Table. DEGs from Δ*eecP* vs OG1RF comparison at T0.**
(XLSX)

**S2 Table. DEGs from Δ*eecP* vs OG1RF comparison at T2.**
(XLSX)

**S3 Table. Chemically defined media formulation.**
(XLSX)

**S4 Table. Primers used in this study.**
(XLSX)

## Acknowledgments

We thank Dr. Joshua Woodward at the University of Washington for the *E. coli* strains expressing DisA and mRECON, Dr. Jeannine Brady at the University of Florida for the COH1 *S. agalactiae* strain, Dr. Nahuel Fittipaldi at the University of Montreal for the NGBS93 *S. agalactiae* strain, Dr. Aria Eshraghi at the University of Florida for the NR-9456 iBMDMs cell line, and Dr. Kari Basso and Katie Heiden at the Mass Spectrometry Research and Education Center at the University of Florida for the mass spectrometry analysis. Mass spectrometry analysis was performed at the Mass Spectrometry Research and Education Center at the University of Florida (NIH S10 OD021758-01A1 and NIH S10 OD030250-01A1).

## Author contributions

**Conceptualization:** Adriana G. Morales Rivera, Anju Bala, Leila G. Casella, Debra N. Brunson, Ana L. Flores-Mireles, José A. Lemos.

**Data curation:** Adriana G. Morales Rivera, Anju Bala, Debra N. Brunson, Ellsa Wongso, Alejandro R. Walker, Shannon M. Wallet.

**Formal analysis:** Adriana G. Morales Rivera, Anju Bala, Leila G. Casella, Debra N. Brunson, Ellsa Wongso, Aria Patel, Ilka E. Cuvilly, Alejandro R. Walker, Shannon M. Wallet.

**Funding acquisition:** José A. Lemos.

**Investigation:** Adriana G. Morales Rivera, Anju Bala, Leila G. Casella, Debra N. Brunson, Ellsa Wongso, Aria Patel, Ilka E. Cuvilly.

**Methodology:** Adriana G. Morales Rivera, Anju Bala, Leila G. Casella, Debra N. Brunson, Ellsa Wongso, Aria Patel, Ilka E. Cuvilly, Shannon M. Wallet, Ana L. Flores-Mireles, José A. Lemos.

**Project administration:** Adriana G. Morales Rivera, José A. Lemos.

**Resources:** Adriana G. Morales Rivera, Anju Bala, Leila G. Casella, Debra N. Brunson, Ellsa Wongso, Aria Patel, Ilka E. Cuvilly, Alejandro R. Walker, Shannon M. Wallet, Ana L. Flores-Mireles, José A. Lemos.

**Supervision:** Shannon M. Wallet, Ana L. Flores-Mireles, José A. Lemos.

**Validation:** Adriana G. Morales Rivera, Anju Bala, Leila G. Casella, Debra N. Brunson, Ellsa Wongso, Alejandro R. Walker, Shannon M. Wallet, Ana L. Flores-Mireles, José A. Lemos.

**Visualization:** Adriana G. Morales Rivera, Anju Bala, Leila G. Casella, Debra N. Brunson, Ellsa Wongso, Aria Patel, Ilka E. Cuvilly, Alejandro R. Walker, Shannon M. Wallet, Ana L. Flores-Mireles, José A. Lemos.

**Writing – original draft:** Adriana G. Morales Rivera, Anju Bala, Debra N. Brunson, Ellsa Wongso, Ana L. Flores-Mireles, José A. Lemos.

**Writing – review & editing:** Adriana G. Morales Rivera, Anju Bala, Debra N. Brunson, Ellsa Wongso, Alejandro R. Walker, Shannon M. Wallet, Ana L. Flores-Mireles, José A. Lemos.

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
