## [Decision Letter · Decision Letter 0]

3 Sep 2025

PPATHOGENS-D-25-01590

Characterization of a novel cell wall-associated nucleotidase of Enterococcus faecalis that degrades extracellular c-di-AMP

PLOS Pathogens

Dear Dr.Lemos,

Thank you for submitting your manuscript to PLOS Pathogens. After careful consideration, we feel that it has merit but does not fully meet PLOS Pathogens's publication criteria as it currently stands. Therefore, we invite you to submit a revised version of the manuscript that addresses the points raised during the review process.

Please submit your revised manuscript within 60 days. If you will need more time than this to complete your revisions, please reply to this message or contact the journal office at plospathogens@plos.org. Please include the following items when submitting your revised manuscript:

We look forward to receiving your revised manuscript.

Kind regards,

Hui Wu

Academic Editor

PLOS Pathogens

Michael Wessels

Section Editor

PLOS Pathogens

Sumita Bhaduri-McIntosh

Editor-in-Chief

PLOS Pathogens

orcid.org/0000-0003-2946-9497

Michael Malim

Editor-in-Chief

PLOS Pathogens

orcid.org/0000-0002-7699-2064

**Additional Editor Comments:**

Please include a detailed, point-by-point response to the following comments and other comments from both reviewers and a tracked-changes manuscript.

1) Link to cGAS–STING signaling. Quantify downstream STING pathway activation during infection with WT, ΔeecP, complemented, and catalytic-His mutant strains (e.g., p-TBK1, p-IRF3, IFN-β mRNA/protein). Establish dependence using cGAS-/- or STING-/- cells (or specific inhibitors) and, where feasible, test at least one host phenotype from Fig. 7D and/or Fig. 8 in cGAS or STING loss-of-function models. Measure extracellular c-di-AMP in the same supernatants/fluids to correlate signal and ligand.

2) Substrate specificity and products. Because TLC suggests activity on pApA and AMP, provide kinetic parameters (Km, kcat) for c-di-AMP vs. alternative substrates under physiologic conditions and identify products by LC–MS. If EecP is multispecific, temper claims attributing phenotypes solely to c-di-AMP and discuss alternatives; include controls (e.g., apyrase/nucleotidase treatments) to test substrate contribution.

3) Attribution to EecP per se. Confirm surface exposure/cell-wall anchoring (fractionation or protease shaving with controls). Present complementation and catalytic mutant data across key assays, and address possible polar/pleiotropic effects.

4) Niche-specific virulence claims. Re-analyze Figs. 7–8 incorporating the above controls; report effect sizes, randomization/blinding, and power justification, and align conclusions to tissues where effects are (or are not) observed.

**Journal Requirements:**

At this stage, the following Authors/Authors require contributions: Adriana G Morales Rivera, Anju Bala, Leila G Casella, Debra N Brunson, Aria Patel, Ellsa Wongso, Ana L Flores-Mireles, and José A Lemos. Please ensure that the full contributions of each author are acknowledged in the "Add/Edit/Remove Authors" section of our submission form.

2) We have noticed that you have uploaded Supporting Information files, but you have not included a list of legends. Please add a full list of legends for your Supporting Information files after the references list.

3) Please ensure that the funders and grant numbers match between the Financial Disclosure field and the Funding Information tab in your submission form. Note that the funders must be provided in the same order in both places as well.

State what role the funders took in the study. If the funders had no role in your study, please state: "The funders had no role in study design, data collection and analysis, decision to publish, or preparation of the manuscript.".

**Reviewers' Comments:**

Reviewer's Responses to Questions

**Part I - Summary**

Reviewer #1: In the manuscript submitted by Rivera, et al., the authors provided solid evidence that an ectonucleotidase in Enterococcus faecalis, EecP, degrades c-di-AMP and generates Pi. They demonstrated that deletion of eecP elevates extracellular c-di-AMP levels. This mutant also exhibited interesting phenotypes during infection in vitro and in vivo. The role of E. faecalis EecP on bacterial pathogenesis is novel.

Reviewer #2: In this manuscript, Rivera et al characterize a novel extracellular cell wall anchored phosphodiesterase in Enterococcus faecalis, EecP, which they discovered through in silico analysis. Unique from many previously studied bacterial ePDEs, EecP contains a duplicated MP-NT two-domain structure. The authors found that EecP degrades extracellular cyclic-di-AMP and that this enzymatic activity is dependent on a catalytic histidine residue. Further corroborating this function, they observed increased levels of extracellular cyclic-di-AMP upon deletion of eecP compared to the parental strain. While deletion of eecP did not impact E. faecalis physiology, it did alter E. faecalis virulence—but, interestingly, in some niches but not others (which had similarly been observed by another bacterial ePDE). The data presented are strong and the manuscript is well-written and clear. Mild concerns exist that no experiments were included to 1) evaluate how changes in extracellular cyclic-di-AMP levels between parental and eecP mutant strains alter innate immune signaling, nor to 2) assess contribution of the cGAS/STING pathway to the pathogenesis differences between the parental and eecP mutant strains. Additional minor concerns are listed below.

**Part II – Major Issues: Key Experiments Required for Acceptance**

Reviewer #1: Apparently, c-di-AMP is not the sole substrate of EecP, as it degrades pApA and AMP as well based on the TLC results. It is known that ectonucleotidases metabolize nucleotides to nucleosides. The substrate of EecP might not limit to c-di-AMP and its intermediates. Since c-di-AMP is not the specific substrate of EecP, the results shown in Figs. 7 and 8 may be related to c-di-AMP, but a consequence due to an effect of EecP on other substrates could not be excluded. Therefore, additional evidence is needed to draw the current conclusions.

Reviewer #2: The authors suggest a positive correlation between extracellular c-di-AMP levels and systemic dissemination, but do not evaluate host signaling in response to accumulated cyclic-di-AMP nor the dependence of these phenotypes on cGAS/STING signaling. The authors should measure the ability of parental Ef vs eecP mutant strains to promote downstream STING signaling (phosphorylated TBK1 and IRF3, or IFNβ production). Further, dependence of the host-related phenotypes observed in Figures 7D and/or Figure 8 on cGAS/STING signaling should be assessed in cGAS and/or STING KO animals/cells as these are commercially available.

**Part III – Minor Issues: Editorial and Data Presentation Modifications**

Reviewer #1: 1. L27. “relay signals to stimuli” – relay stimuli to signals?

2. L40 and later. EecP is not a phosphodiesterase based on the data presented.

3. Fig. S2. Where is Pi from if the enzyme cleaves one molecule of c-di-AMP into two molecules of AMP?

4. Fig. 4E. Degradation of c-di-AMP by the ∆eecP mutant, although less dramatic than that by the WT, was not discussed.

Reviewer #2: • It seems somewhat counterintuitive to label EecP as a virulence factor (lines 138-140) given that deletion of eecP enhances dissemination in both animal models of infection. The authors should clarify this rationale.

• Is anything known concerning regulation of eecP (potentially from mining previously published RNA-Seq datasets) as well as the mechanisms by which cyclic-di-AMP is secreted? It would be helpful to add this information to the Introduction and/or Discussion.

• in Fig 1B-C what does the light blue/teal superimposition represent? I did not see these long, disordered regions mentioned in the text. Labeling of residues in these structural models in addition to the color-coding would also be helpful.

• In Fig 4C-D, there appears to be much higher levels of cyclic-di-AMP in the extracellular compartment compared to the intracellular compartment. Is this the case or is this due to lower amounts of total protein in the supernatant compared to the pellet (the denominator is lower, resulting in a higher normalized value).

• Figure 6 and possibly Fig 7A-B should be moved to supplementary material

• eecP not italicized in Fig 7B

• Figure 7D—Is decreased intracellular survival by the eecP mutant here due to decreased initial adherence to or uptake by host cells?

• More experimental details should be included in the Figure 8 legend—for example, n per group, how many independent animal experiments were performed, are the data shown representative of one experiment or were multiple experiments combined, etc. Animal experiments are known to be variable— can the authors address why outliers were removed in Figure 8? These should be added back in to the results.

PLOS authors have the option to publish the peer review history of their article (what does this mean?). If published, this will include your full peer review and any attached files.

Reviewer #1: No

Reviewer #2: No

**Figure resubmission:**
---

## [Editor Report · Decision Letter 1]

12 Apr 2026

PPATHOGENS-D-25-01590R1

Characterization of a novel cell wall-associated nucleotidase of Enterococcus faecalis that degrades extracellular c-di-AMP

PLOS Pathogens

Dear Dr. Lemos,

Thank you for submitting your manuscript to PLOS Pathogens. After careful consideration, we feel that it has merit but does not fully meet PLOS Pathogens's publication criteria as it currently stands. Therefore, we invite you to submit a revised version of the manuscript that addresses the points raised during the review process.

We look forward to receiving your revised manuscript.

Kind regards,

Hui Wu

Academic Editor

PLOS Pathogens

Michael Wessels

Section Editor

PLOS Pathogens

Sumita Bhaduri-McIntosh

Editor-in-Chief

PLOS Pathogens

orcid.org/0000-0003-2946-9497

Michael Malim

Editor-in-Chief

PLOS Pathogens

orcid.org/0000-0002-7699-2064

**Additional Editor Comments:**

The study addresses an interesting and timely question and presents a potentially important advance in understanding extracellular cyclic dinucleotide biology in E. faecalis. The identification of EecP as a surface-associated enzyme influencing host-pathogen interactions is novel, and the revised manuscript is strengthened by additional new data supporting cell-wall association, pharmacologic linkage to cGAS-STING signaling, and expanded immune transcriptional profiling.

That said, several issues remain only partially resolved and should be addressed or discussed before the manuscript can be accepted.

Major points

1. Mechanistic linkage to cGAS-STING signaling remains suggestive rather than definitive.

2. Substrate specificity remains insufficiently defined in the absence of purified protein, kinetic parameters, or product validation means.

3. In vivo conclusions remain cautious. The manuscript should avoid overinterpreting niche-specific virulence effects and should frame these findings more conservatively as context-dependent infection phenotypes.

Minor issues

1. Please ensure that the manuscript consistently describes EecP as an immune-modulating factor rather than a straightforward virulence factor.

2. Some biochemical interpretation remains speculative, including the proposed AMP-to-adenosine/Pi activity and the putative role of the second domain; these points should be clearly labeled as hypotheses rather than conclusions.

3. The discussion would benefit from a clearer statement of what remains unresolved, particularly regarding export of c-di-AMP, the exact host sensor hierarchy, and whether alternative extracellular nucleotides contribute to the observed immune effects.

**Reviewers' Comments:**

**Figure resubmission:**

While revising your submission, we strongly recommend that you use PLOS’s NAAS tool (https://ngplosjournals.pagemajik.ai/artanalysis) to test your figure files. NAAS can convert your figure files to the TIFF file type and meet basic requirements (such as print size, resolution), or provide you with a report on issues that do not meet our requirements and that NAAS cannot fix. After uploading your figures to PLOS’s NAAS tool - https://ngplosjournals.pagemajik.ai/artanalysis, NAAS will process the files provided and display the results in the "Uploaded Files" section of the page as the processing is complete. If the uploaded figures meet our requirements (or NAAS is able to fix the files to meet our requirements), the figure will be marked as "fixed" above. If NAAS is unable to fix the files, a red "failed" label will appear above. When NAAS has confirmed that the figure files meet our requirements, please download the file via the download option, and include these NAAS processed figure files when submitting your revised manuscript.
---

## [Editor Report · Decision Letter 2]

27 Apr 2026

Dear Dr. Lemos,

We are pleased to inform you that your manuscript 'Characterization of a novel cell wall-associated nucleotidase of Enterococcus faecalis that degrades extracellular c-di-AMP' has been provisionally accepted for publication in PLOS Pathogens.

Best regards,

Hui Wu

Academic Editor

PLOS Pathogens

Michael Wessels

Section Editor

PLOS Pathogens

Sumita Bhaduri-McIntosh

Editor-in-Chief

PLOS Pathogens

orcid.org/0000-0003-2946-9497

Michael Malim

Editor-in-Chief

PLOS Pathogens

orcid.org/0000-0002-7699-2064

Thanks for submitting the manuscript and all your efforts to improve the manuscript.
---

## [Editor Report · Acceptance letter]

Dear Dr. Lemos,

We are delighted to inform you that your manuscript, "Characterization of a novel cell wall-associated nucleotidase of Enterococcus faecalis that degrades extracellular c-di-AMP," has been formally accepted for publication in PLOS Pathogens.

Best regards,

Sumita Bhaduri-McIntosh

Editor-in-Chief

PLOS Pathogens

orcid.org/0000-0003-2946-9497

Michael Malim

Editor-in-Chief

PLOS Pathogens

orcid.org/0000-0002-7699-2064